# An Overview of the Recent Advances in Composite Materials and Artificial Intelligence for Hydrogen Storage Vessels Design

Mourad Nachtane [1,*], Mostapha Tarfaoui [2,3], Mohamed amine Abichou [1], Alexandre Vetcher [4,5], Marwane Rouway [2], Abdeouhaed Aâmir [6], Habib Mouadili [7], Houda Laaouidi [2] and Hassan Naanani [8]

1 S Vertical Company, F-92290 Paris, France
2 ENSTA Bretagne, IRDL, UMR CNRS 6027, F-29200 Brest, France
3 Green Energy Park (IRESEN/UM6P), Benguerir 43150, Morocco
4 Institute of Biochemical Technology and Nanotechnology (IBTN), Peoples' Friendship University of Russia (RUDN), 6 Miklukho-Maklaya St., 117198 Moscow, Russia
5 Complementary and Integrative Health Clinic of Dr. Shishonin, 5 Yasnogorskaya St., 117588 Moscow, Russia
6 Laboratory of Mechanical Energy Electronic Telecommunications (MEET), Faculty of Science and Technology, Hassan I University, Casa Road, Settat B.P. Box 577, Morocco
7 Equipe I3MP, Laboratoire (GeMEV), Faculté des Sciences Aïn Chock Maârif, Casablanca B.P. Box 5366, Morocco
8 Laboratory of Inorganic Materials for Sustainable Energy Technology-LIMSET, UM6P, Benguerir 43150, Morocco; hassan.naanani@um6p.ma
* Correspondence: mourad.nachtane@ensta-bretagne.org; Tel.: +33-699673572

**Abstract:** The environmental impact of $CO_2$ emissions is widely acknowledged, making the development of alternative propulsion systems a priority. Hydrogen is a potential candidate to replace fossil fuels for transport applications, with three technologies considered for the onboard storage of hydrogen: storage in the form of a compressed gas, storage as a cryogenic liquid, and storage as a solid. These technologies are now competing to meet the requirements of vehicle manufacturers; each has its own unique challenges that must be understood to direct future research and development efforts. This paper reviews technological developments for Hydrogen Storage Vessel (HSV) designs, including their technical performance, manufacturing costs, safety, and environmental impact. More specifically, an up-to-date review of fiber-reinforced polymer composite HSVs was explored, including the end-of-life recycling options. A review of current numerical models for HSVs was conducted, including the use of artificial intelligence techniques to assess the performance of composite HSVs, leading to more sophisticated designs for achieving a more sustainable future.

**Keywords:** hydrogen storage; composite materials; transport applications; circular economy; recycling; artificial intelligence; sustainable development

## 1. Introduction

Fossil fuels threaten the environment because they use natural resources, make greenhouse gases worse, and pollute the air. Cleaner energy is required [1–4], and hydrogen is a possible solution for transport applications. In addition to enabling the decarbonization of road transportation, hydrogen energy can also significantly reduce local air pollution [5]. Hydrogen is a non-toxic, colorless, and odorless gas with the highest energy density per mass compared to standard fuels, and more crucially, its fuel infrastructure is on par with traditional road fuels [6].

In the past few years, there has been a rise in the number of programs to make hydrogen meet energy and climate goals. Since 2000 [7], 230 projects have been launched worldwide to convert electrical energy to hydrogen (Figure 1). The capital costs of water electrolyzers installed in 2017 and 2018 are about USD 20–30 million annually. Additional investments in storage tanks, refueling infrastructure, pipes, and other equipment will make the total cost of a project even higher. Both alkaline and proton exchange membrane

(PEM) electrolyzers are routinely used in these programs, but recent projects have tended to favor PEM, reflecting that many test conditions favor less mature technologies with considerable cost-saving potential.

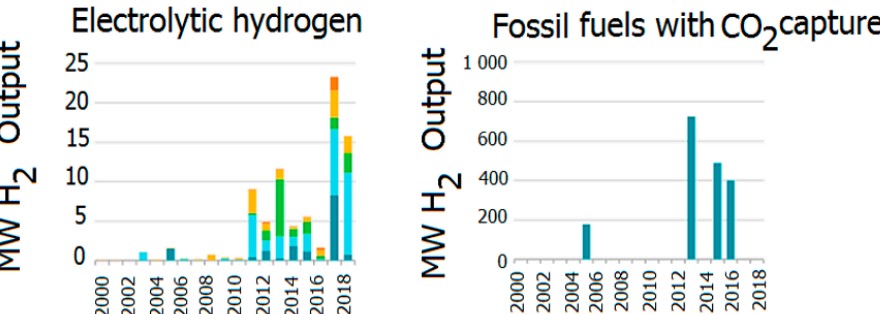

**Figure 1.** Hydrogen production capacity by technology and start date [7].

Solid oxide electrolyzer cells, which promise improved efficiency, are also entering this market. So far, the electrolyzers in these projects have not been more significant than 10 megawatts (MWe) (with modules between 2 and 4 MW), and most of them have been much smaller [8]. A 20 MWE project is now under development, while several project ideas exceed the 100 MWE mark. Several programs have demonstrated hydrogen conversion to synthetic methane, methanol, ammonia, and other hydrogen-based fuels and feedstocks [9].

Hydrogen, however, can be stored in various physical and chemical ways. Three techniques for onboard hydrogen storage in transport vehicles have been considered: storage in compressed gas (35 MPa or 70 MPa), storage as cryogenic liquid (20 K), and solid storage [10]. It is possible to distinguish three prominent families of materials enabling the solid storage of hydrogen: reversible metal hydrides ($LaNi_5H_6$, $FeTiH_2$, $Mg_2NiH_4$ ... ), complex hydrides ($NaAlH_4$, $LiAlH_4$, $NaBH_4$ ... ), and porous materials such as nanotubes of carbon and zeolites [11]. These technologies are now in competition to meet the requirements of vehicle manufacturers.

Materials science and research on new materials have made it possible for engineers to think about using less common materials in their designs [12–14]. Composite materials are gradually being used for more applications in marine, aerospace, automotive, and other industries [15–18]. Composites offer excellent strength-to-weight ratios, improved thermal and mechanical properties, and many other desirable aspects obtained by combining the different constituent materials [19–23]. Their use in hydrogen storage vessels could increase the efficiency of the system. Hydrogen storage tank design must consider the application, test pressure, the external environment (including potential mechanical effects, chemical degradation, integration, etc.), life cycle, and safety factors defined for stationary and portable applications [24]. Failure modes and operating conditions should also be considered when selecting materials [25–27].

Hydrogen vessels often use composite materials as the most mature solution. These tanks are typically made using filament winding processes, which depend highly on the choice of materials. Therefore, researchers have since explored multiple material options for resin matrix systems and liner systems [28]. However, several limiting factors prevent the broader use of composites for producing sophisticated hydrogen storage tanks, including high life cycle costs and poor recyclability. Depending on the tank size and production volume, carbon fiber accounts for approximately 45–80% of the tank cost [29]. Furthermore, the environmental impact of hydrogen mobility should be assessed and compared to that of other modes of transportation, including the recycling of hydrogen system components at the end of their useful lives (fuel cells and tanks, in particular) [30]. To address these issues, thermoplastics appear to offer a promising solution regarding hydrogen compatibility and the demands of mass automotive markets [31]. Nanocomposites have piqued

the interest of researchers due to their superior mechanical, electrical, electronic, optical, magnetic, and surface properties [32]. These materials' high surface area/volume ratio has crucial implications for energy storage. The enormous surface area and the possibility for nanomaterial improvement are essential characteristics of this new class of hydrogen storage materials. Several research investigations have demonstrated the acute effects of nanofillers on tensile characteristics, thermal conductivity, thermal stability, and gas barrier qualities [33,34].

This paper examines using composites and nanocomposites in hydrogen storage tank applications. Different machine learning techniques are applied to characterize and predict the performance of composite materials at different length scales. In practice, these methods use dimensionality reduction techniques to reduce the generated dataset and train a machine learning classifier to predict mechanical behavior at the structural level [35,36]. This paper presents the application of artificial intelligence and digital twin technologies for predicting the hygro-thermomechanical behavior of hydrogen storage vessels. The article is structured as follows: Section 2 provides an overview of current hydrogen storage vessel types. The hydrogen storage vessel design is explored in Section 3. The use of composite materials and nanocomposites for hydrogen storage vessels is presented in Section 4. Section 5 comprehensively reviews the numerical approaches employed in modeling the hygro-thermomechanical behavior of hydrogen storage vessels. The artificial intelligence and optimization models for modeling the composite hydrogen storage tanks are presented in Section 6.

## 2. Different Types of Hydrogen Storage

Current options for onboard hydrogen storage include compressed hydrogen gas, cryogenic and liquid hydrogen, sorbents, metal hydrides, and chemical hydrides (Figure 2) [37].

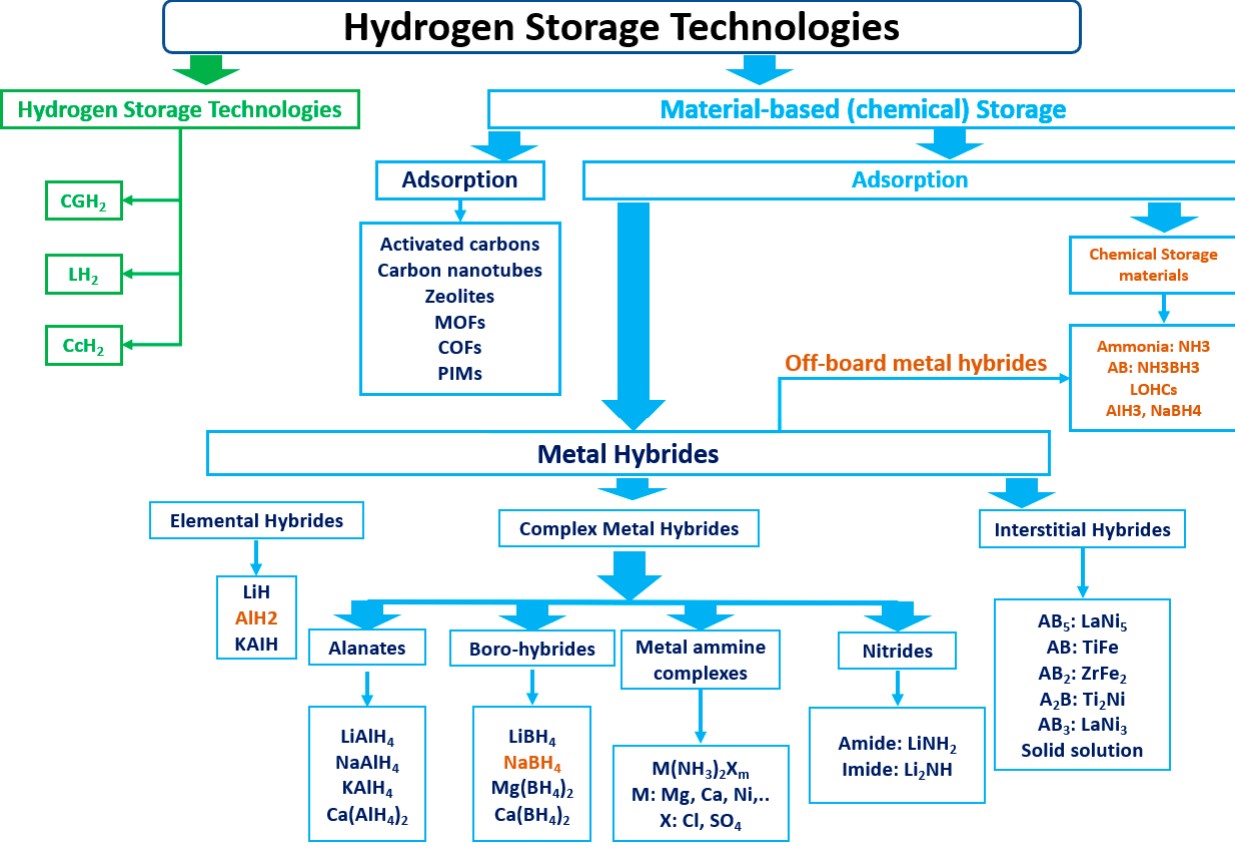

**Figure 2.** Different types of hydrogen storage [37].

### 2.1. Physical Storage

2.1.1. Compressed Hydrogen Storage (CGH$_2$)

The most established hydrogen storage technology is compressed gas storage in high-pressure tanks. There are four conventional forms of hydrogen storage vessels: Type I, Type II, Type III, and Type IV (see Figure 3) [38]. Type I vessels are all-metal (generally steel) and thus the heaviest; they are primarily used in industry for fixed use. At 200–300 bar, Type I vessels store around 1% of hydrogen globally [39]. Type II vessels have a composite sleeve (hoop direction) over a metal liner that weighs less than Type I. Both Type I and II vessels are unsuitable for vehicle applications due to their low hydrogen storage density, caused by high mass and hydrogen embrittlement issues. A fully wrapped composite cylinder with a metal liner that functions as a hydrogen permeation barrier is used in Type III vessels. The metal liner is generally aluminum (Al), eliminating embrittlement and providing more than 5% mechanical resistance [40]. Type III vessels have a mass increase of 25–75% over Types I and II vessels, making them more suited for vehicle applications; nevertheless, they are more expensive. Type III vessels have also been proven reliable at pressures up to 450 bar; however, pressure cycling tests over 700 bar continue to provide issues [41]. A fully wrapped composite cylinder with a plastic liner (usually high-density polyethylene) that serves solely as a hydrogen permeation barrier is used in Type IV vessels. The load-bearing structure is the composite overwrap, commonly comprised of carbon fiber or carbon/glass fiber composite in an epoxy matrix. Type IV pressure vessels are the lightest of the pressure vessels, making them ideal for vehicle applications, and they can withstand pressures of up to 1000 bar [42]. However, they are prohibitively expensive because of the high cost of carbon fibers. Considering the high production volumes, cost estimations suggest that carbon fiber accounts for approximately 75% of the storage vessel cost. Type V tanks are at the design stage and are likely to use a thermoplastic liner and composite structure that are more closely linked; the composite and the liner will be made from the same thermoplastic polymer [43], but further development is required to ensure these tanks are safe in service.

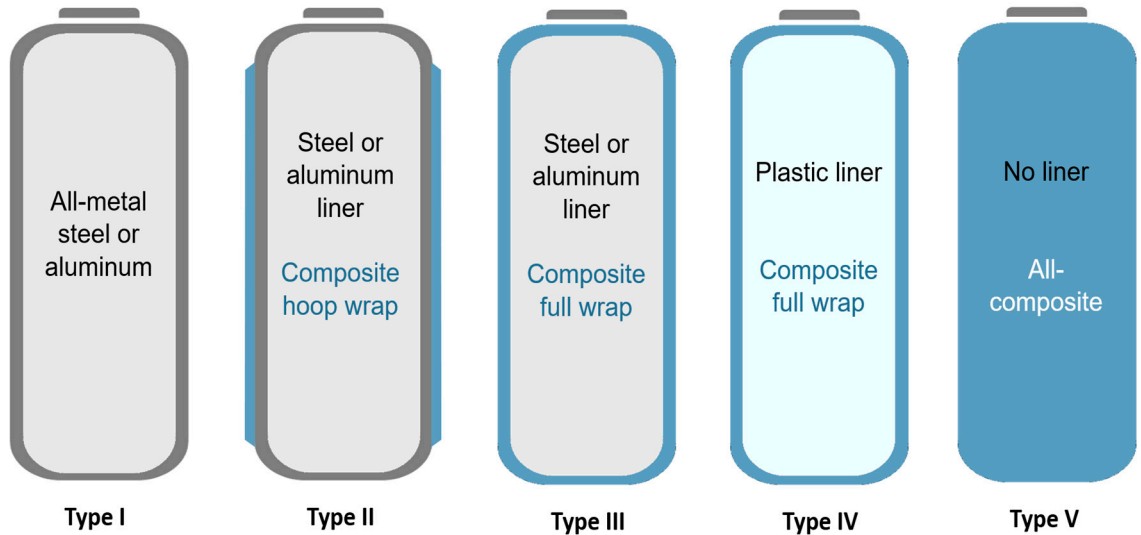

**Figure 3.** Schematic depiction of the five types of pressure vessels [38].

The cost is a critical factor when considering the different types of hydrogen storage vessels. Compressed gas storage is generally the most cost-effective option, with a relatively low cost per kilogram of hydrogen storage capacity. However, the tanks can be heavy and bulky, and frequent refilling is required. Liquid hydrogen storage is a more expensive option, with higher upfront costs for the specialized tanks and insulation required to maintain low temperatures. Metal hydride and chemical hydride storage are newer technologies with higher storage densities and safety, but they can be significantly more expensive than compressed gas or liquid hydrogen storage. Ultimately, the choice of hydrogen storage

technology will depend on several factors, including the application, required storage capacity, and cost-effectiveness. Table 1 presents the different types of hydrogen storage tanks and their characteristics.

**Table 1.** Characteristics of different types of hydrogen storage tanks.

| Hydrogen Storage Tank Type | Energy Density (MJ/kg) | Operating Pressure (bar) | Temperature Range (°C) | Cost ($/kg H2) | Refill Time (min) |
|---|---|---|---|---|---|
| Type I | 4–5 | 250–700 | −40 to 65 | $7–10 | 5–10 |
| Type II | 4–5 | 700–875 | −40 to 65 | $5–8 | 3–5 |
| Type III | 4–5 | 875–1100 | −40 to 65 | $4–6 | 2–3 |
| Type IV | 4–5 | 700–875 | −40 to 65 | $7–10 | 3–5 |
| Type V | 5–8 | 875–1100 | −40 to 65 | $5–7 | 2–3 |

Note: These values are approximate and can vary depending on various factors such as the specific design, manufacturing process, and materials used in the hydrogen storage tank. Additionally, different sources may report slightly different values.

### 2.1.2. Cryogenic Hydrogen Storage (LH$_2$)

Hydrogen can also be stored as a cryogenic liquid at atmospheric pressure and in compressed hydrogen. Cryogenic hydrogen storage is better than compressed hydrogen gas storage because it is safer and takes up less space. When hydrogen is stored as a liquid, its volumetric density goes up. At the boiling point of hydrogen (253 °C) and atmospheric pressure, the theoretical volumetric density of liquid hydrogen (LH$_2$) is 70 g L1, while it is 24 g L1 and 40 g L1 for compressed hydrogen at 350 bar and 700 bar, respectively, at room temperature [44]. The main problem with using liquefaction is that it requires more energy to create the liquid than to compress a gas. Hydrogen has a critical temperature of −240 °C, above which it cannot condense. Since hydrogen has a low boiling point, it can only be stored as a liquid at low temperatures. Therefore, expensive cooling must be used. This energy-intensive process uses 25–40% of the energy in the hydrogen, compared with using only 10% of the energy to compress it as a gas [45]. LH$_2$ cryogenic vessels, therefore, must be vacuum insulated to maintain such low temperatures. They typically have two walls and a vacuum between them to prevent heat from escaping. However, boil-off losses cannot be avoided since the heat from the environment flows into the LH$_2$ and moves through other parts, which can be as high as 0.4% per day [46]. When making vessels for cryogenic hydrogen, one of the main goals is to ensure the surface area of the liquid is as tiny as possible, which will prevent heat from entering the liquid from the atmosphere.

Boil-off losses can be dangerous if the vessel is in a small space. For example, if a car with a cryogenic hydrogen tank is left in a closed garage for a few days, the hydrogen could boil off and cause a fire. BMW has developed vehicles with internal combustion engines that can run on gasoline and cryogenic hydrogen [47]. There are also space applications and medium- to large-scale transportation examples that use liquid hydrogen. In addition to LH$_2$ trailers that can hold about 4000 kg of H$_2$ [48], LH$_2$ ships can also transport H$_2$ worldwide. Kawasaki Heavy Industries Ltd. is currently making LH$_2$ ships that can hold up to 11,500 tons (160,000 m$^3$) of H$_2$ [44].

### 2.1.3. Cryo-Compressed Hydrogen Storage (CcH$_2$)

Cryo-compressed hydrogen storage combines the benefits of both compressed and cryogenic hydrogen storage. As previously noted, one of the drawbacks of compressed hydrogen storage is the enormous volume and high pressures required. Furthermore, the unavoidable boil-off losses are one of the drawbacks of cryogenic hydrogen storage. Cryo-compressed storage alleviates all of these issues. In this case, the insulated vessel that holds hydrogen can resist cold temperatures and high pressures, increasing the volumetric hydrogen storage capacity and safety of compressed hydrogen or cryogenic LH$_2$. Compressing liquefied hydrogen at 20 K improves the volumetric hydrogen storage capacity from 70 g L1 at 1 bar to 87 g L1 at 240 bar [49]. The ability of the insulated vessel to withstand high

pressures allows for a higher rise in pressure within the tank than in cryogenic storage, as well as extended dormancy periods, resulting in enhanced storage density and lower boil-off losses. Furthermore, the lower pressures used in cryo-compressed hydrogen storage (usually 300 bar) versus compressed hydrogen storage (700 bar) may lessen the need for more expensive carbon fiber composites. A technical assessment of a cryo-compressed hydrogen storage vessel revealed that this option could meet specific US DOE system targets for automotive applications, including gravimetric and volumetric hydrogen capacities and hydrogen loss during dormancy under minimum daily driving conditions [50]. The cryo-compressed hydrogen storage system was expected to improve gravimetric and volumetric capacities by 91% and 175%, respectively, with a 46% reduction in carbon fiber composite mass and a 21% reduction in system cost. Furthermore, a dormancy time of more than seven days without a loss was found, with an initial tank filling at 85%. A prototype cryo-compressed hydrogen vehicle was presented in 2012 [51], but the availability and cost of the infrastructure remain significant barriers to this storage option, limiting its viability.

### 2.2. Chemical Storage

Chemical storage of hydrogen may offer higher energy densities and the possibility of being easier to use by the general public. The storage material would have to be regenerated away from the vehicle because it cannot be created by subjecting the hydrogen gas to high pressures at ambient temperatures. Several chemical systems that release hydrogen in both exothermic and endothermic ways are currently being studied. Chemical compounds that contain hydrogen can also be thought of as a way to store hydrogen. Methanol, ammonia, and methylcyclohexane are some examples. Under STP conditions, all these compounds are liquids, so the same infrastructure that moves and stores gasoline could also be used for these compounds; this is a clear advantage compared to gaseous hydrogen, which needs pipes and vessels that do not leak and are preferably seamless. These chemical compounds can hold much more hydrogen [52]: 8.9 wt.% for methanol, 15.1 wt.% for ammonia, and 13.2 wt.% for methylcyclohexane. Several attempts have been made to derive hydrogen from ammonia boreane (borazane) when it is solid and dissolved in water [53,54]. Catalysts, such as a variety of acids and complexes of transition metals, have been found and are being modified to increase both the amount of hydrogen released and the rate at which it is released [55]. However, ammonia-borane can only work as an onboard hydrogen storage material if the fuel used for dehydrogenating ammonia-borane can be recycled in a way that is both efficient and inexpensive. However, the storage method is irreversible, and the compounds cannot be charged with hydrogen repeatedly. The compounds must be made in a central plant, and the reaction waste must be reused. This is hard to do, especially with ammonia, which makes nitrogen oxides bad for the environment. Carbon oxides, which are also bad, are made by other compounds (Figure 4).

Renewable hydrogen, produced using 100% renewable electricity for water electrolysis, is a near-zero greenhouse gas (GHG) energy source that the European Union (EU) could utilize in its efforts to decarbonize. Hydrogen would need to be supplied to a hydrogen refueling station when used for transportation (HRS). Hydrogen can be produced at a central location and transported to the HRS, or it can be produced directly at the HRS. Unlike centralized production, onsite hydrogen production eliminates costly and inefficient fuel transportation from the production site to the HRS. In this study, we investigated the current and future price of renewable hydrogen at the pump in EU countries using HRSs with onsite electrolysis.

Figure 5 depicts the average EU pump price for onsite renewable hydrogen, its breakdown into hydrogen production (teal bar) and fueling cost (orange bar), and the total cost when a 3-euro-per-kilogram subsidy is applied (grey diamond). Using the median scenario and a 30% HRS utilization rate, we estimate that the average pump price of renewable hydrogen in the EU in 2020 will be 11 euros per kilogram. We anticipate that the costs of producing renewable hydrogen will decrease in the future as a result of technological advances in both renewable electricity generation and electrolysis, including likely cost

reductions in electrolyzers. We anticipate that the levelized cost of HRS infrastructure will decrease on a per-kilogram basis, primarily as a result of increased utilization rates, which we assume to be 50% in 2030 and 70% in 2050. Regardless of these cost reduction assumptions, even our most optimistic estimate of 6 euros per kilogram of hydrogen is significantly higher than the European Commission president's target of 1.8 euros per kilogram for 2030.

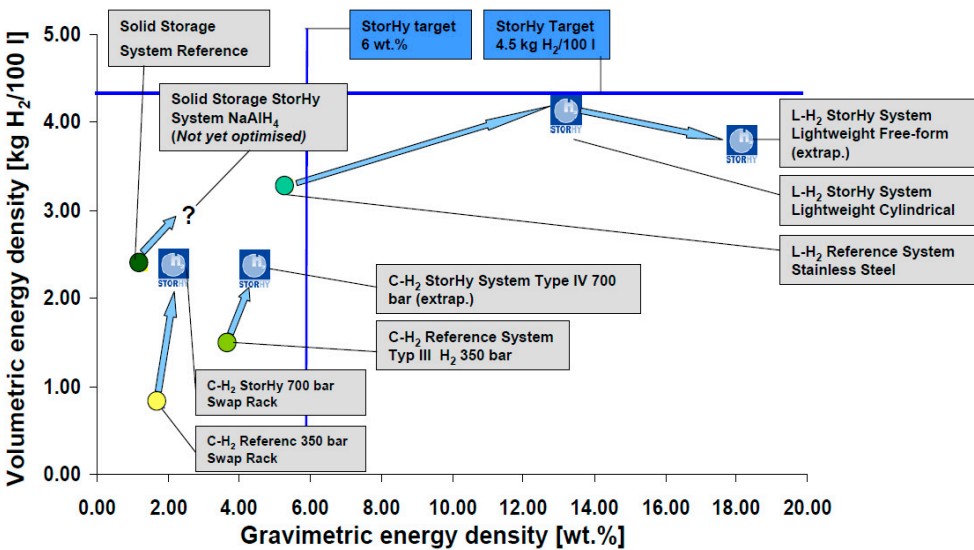

**Figure 4.** Comparison of system storage densities [56].

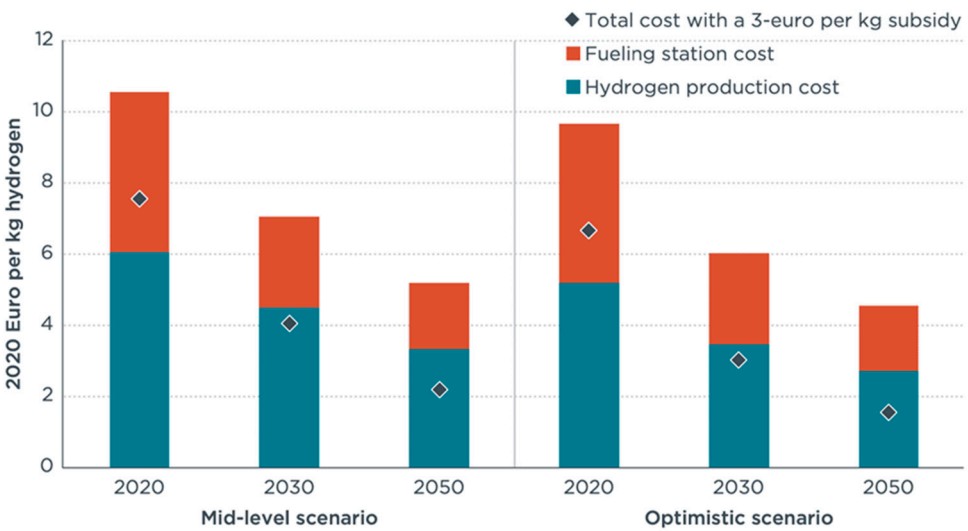

**Figure 5.** The average price of hydrogen at the pump in 26 EU countries, based on the mid-level cost scenario and the optimistic cost scenario [57].

## 3. Hydrogen Storage Vessels Design

Pressure vessel engineering design approaches have changed throughout time, from approximate calculations using classical laminate theory to thorough finite-element analysis methods that analyze precisely the stresses and strains during various internal and external load circumstances [58]. In order to maximize the amount of $H_2$ that can be stored while staying within the allowed volume range, weight, and maximum operating pressure, hydrogen storage tanks need to be made using complex methods. Material selection is an essential step in the design process. Materials must be hydrogen compatible within the projected boundary conditions, which include exposure level (pressure), mechanical stress levels, temperature

extremes, and fatigue loading. Under onboard operation, materials that function well under static load at room temperature may become sensitive to hydrogen embrittlement and time-dependent crack propagation. The reaction to moisture-contaminated hydrogen requires careful study, especially at high temperatures. Hydrogen seals must work reliably at severe temperatures and under numerous pressure cycles and dynamic stresses.

When developing hydrogen storage tanks, five major assessment areas must be considered (Figure 6) [59]. The technical performance of HSVs is a critical factor in determining their effectiveness as a hydrogen storage solution. Several key design elements impact the technical performance of HSVs, including the materials used, the pressure rating, and the size and shape of the vessel. The materials used in the construction of HSVs play a significant role in determining their performance. Steel and aluminum are common materials used in HSV construction due to their strength and durability. However, composite materials, such as fiber-reinforced plastics, are increasingly being used due to their lighter weight and improved resistance to corrosion [60]. The pressure rating of an HSV refers to the maximum pressure that the vessel can safely contain. High-pressure HSVs typically have a pressure rating of 700 bar or more, while low-pressure HSVs have a rating of 350 bar or less. The pressure rating of an HSV will impact its overall storage capacity and efficiency. The size and shape of an HSV can also impact its technical performance. HSVs come in various shapes, including spherical, cylindrical, and toroidal. The shape of an HSV can impact its hydrogen storage capacity and stability, with some shapes providing improved performance over others. Therefore, the cost of manufacturing HSVs can vary greatly depending on the materials used, the design complexity, and the manufacturing process. Steel and aluminum HSVs tend to be less expensive to manufacture than composite HSVs.

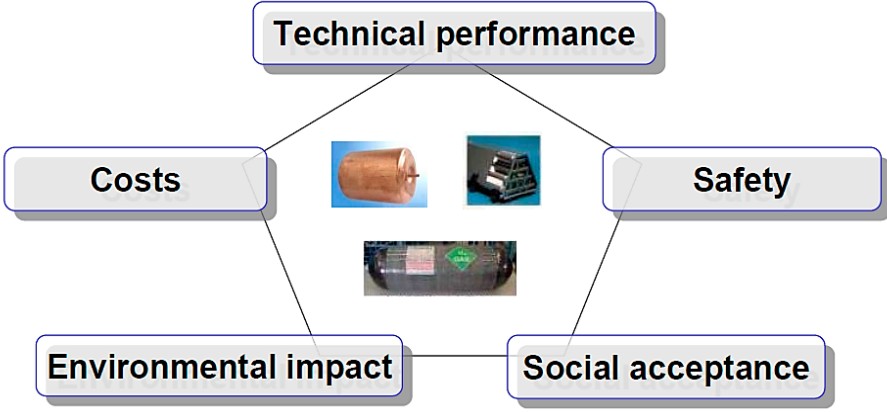

**Figure 6.** Multi-criteria evaluation for hydrogen storage tank design [59].

However, composite HSVs often offer improved performance and reduced weight compared to metal-based vessels. Mass production and advanced manufacturing techniques can also help reduce the cost of HSV production. Also, safety is a critical aspect of HSV design, and HSVs must meet strict safety and regulatory standards, such as those set by the Department of Transportation and the International Organization for Standardization. For example, high-pressure gas cylinders must be designed to withstand high pressures, temperatures, and impact, and they must be tested to ensure they meet these requirements. Finally, the environmental impact of Hydrogen Storage Vessel (HSV) designs can be reduced by using recyclable materials in their production. For example, high-pressure gas cylinders can be made from recycled steel or aluminum, reducing the emissions associated with producing new materials and conserving finite resources. Composite overwrapped pressure vessels (COPVs) can also be made using recyclable materials, such as carbon fiber, which can be recovered and reused after the end of the HSV's life. Additionally, using recyclable materials to produce COPVs can reduce the emissions associated with producing new materials and conserve finite resources [61].

### 3.1. Design Procedure

Several integrated characteristics, including progressive failure qualities, burst pressure, and fatigue lifetime, are incorporated into the design of a composite vessel. Creating a composite vessel that combines high reliability with practicality is challenging from the outset of the design process. From a micromechanics point of view, composite failure is complicated, including matrix cracking, fiber/matrix debonding, delamination, fiber rupture, and interactions between these modes. Optimization can be used to make composite vessels lighter, stronger, and more reliable by designing the orientations and ply thicknesses of the wound fibers. Classical laminate theory (CLT), thick cylinder theory, or netting analysis [42] can be used to determine the preliminary sizes of the composite layup for hydrogen tanks (Figure 7). CLT is used to evaluate each layer's layup sequence and thickness. Some researchers have used the netting analysis [62], which uses the principle of static equilibrium and assumes that all fibers are loaded in tension and do not have any shearing or bending stresses. According to grid theory, the basic assumptions are that: (1) only the longitudinal carbon fiber bear the pressure, and (2) the effects of wound patterns are neglected. Grid theory can calculate each composite layer's thickness and the longitudinal in situ fiber strength. The burst pressure of the composite vessel inversely determines the strength value. The analytical method for studying the mechanical response of thick-walled composite pipes subjected to internal pressure was proposed by Bouhafs et al. (2012) [63].

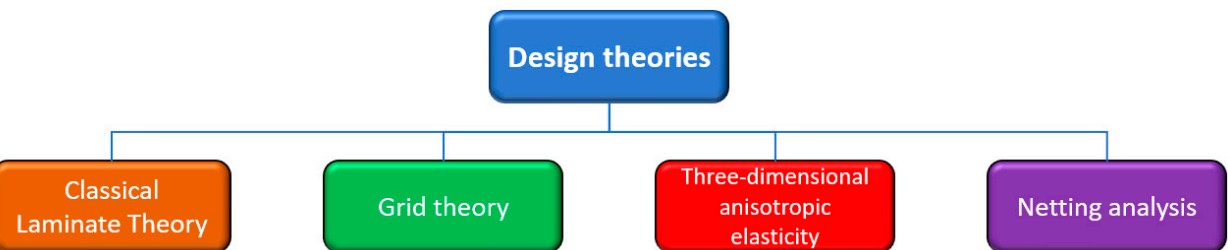

**Figure 7.** Design theories of hydrogen storage tanks.

Some problems with making hydrogen tanks for commercially available fuel cell vehicles include reducing the filament winding cycle time and making sure they are safe, light, cheap, etc. The US Department of Energy's (DOE) Office of Fuel Cell Technologies saw these problems and started a project to design and build hydrogen tanks. The project looked into different options for the matrix resin, carbon fiber, and the shape of the tank design [64]. However, for the resin option, they did not think about thermoplastic resin, which is good for the environment and shortens the time it takes to make something. The designer must take into account the general characteristics, such as tank capacity, working pressure, material attributes, and safety factors. A complete design cycle of a composite hydrogen tank is illustrated in Figure 8.

### 3.2. Numerical Analysis of Composite Hydrogen Storage Vessels

Composite hydrogen storage vessels are subjected to complex loading conditions during their operation. These loading conditions include mechanical loads, such as pressure, tension, and shear, and thermal loads, such as temperature fluctuations. The composite materials used in these vessels are also subjected to cyclic loading due to the filling and emptying of the vessel. The complex loading conditions experienced by composite hydrogen storage vessels can significantly impact their structural and thermal performance [60]. Similarly, temperature fluctuations can cause thermal stresses and deformations within the vessel, leading to failure if the vessel is not designed to withstand these loads. It is essential to accurately model and analyze these loading conditions to ensure the safe and efficient operation of composite hydrogen storage vessels under complex loading conditions. This can be done through numerical analysis techniques, such as finite element analysis (FEA), to predict the structural and thermal behavior of the vessel under different loading conditions [65]. The results of these analyses can be used to optimize the vessel's

design and identify any potential failure modes or weaknesses in the vessel's design. The numerical analysis of composite hydrogen storage vessels involves using mathematical models and computational techniques to analyze these vessels' structural and thermal performance. The main objective of this analysis is to evaluate the safe and efficient operation of these vessels under various loading conditions, such as pressure, temperature, and cyclic loading [42]. To perform the numerical analysis, the first step is to define the geometry and material properties of the vessel. This includes the thickness and type of composite material used, as well as the size and shape of the vessel. The material properties, such as Young's modulus, Poisson's ratio, and the thermal expansion coefficient, are essential for predicting the mechanical behavior of the vessel under different loading conditions. Next, the structural and thermal loads acting on the vessel need to be identified. These loads can include internal pressure, temperature fluctuations, and cyclic loading due to the filling and emptying of the vessel [66]. The boundary conditions of the vessel, such as the temperature of the surrounding environment and the support conditions, also need to be defined. Once the vessel geometry, material properties, and loading conditions have been defined, a finite element model can be created to analyze the structural and thermal performance of the vessel. This model can predict the vessel's deformation, stress, and strain distribution under different loading conditions. It can also be used to predict the temperature distribution within the vessel, which is vital for evaluating the thermal stability of the vessel. The numerical analysis of composite hydrogen storage vessels can provide valuable insights into the performance and safety of these vessels, helping to ensure their safe and efficient operation.

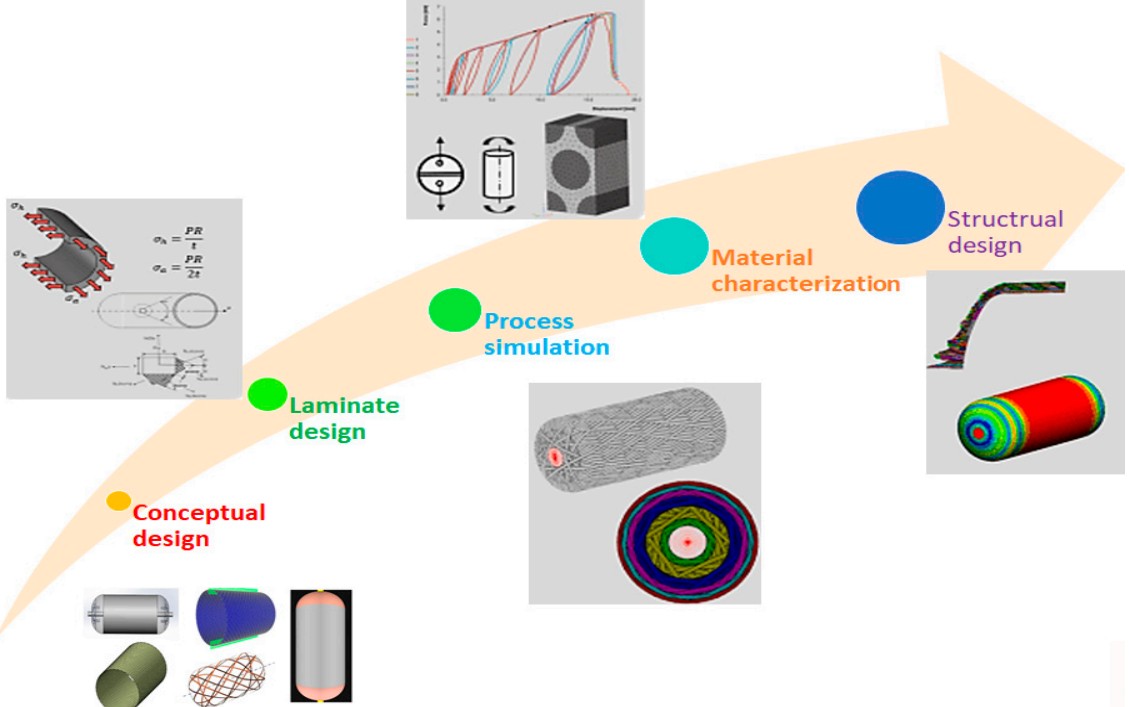

**Figure 8.** Design methodology of hydrogen storage vessels.

## 4. Composite Materials for Hydrogen Storage Vessels

Composite materials have been widely studied for use in hydrogen storage vessels due to their high strength-to-weight ratio and excellent resistance to corrosion and fatigue. These materials allow for a lightweight and durable structure that can withstand the high pressure and temperature requirements of hydrogen storage [67]. One of the main benefits of using composite materials for hydrogen storage vessels is their ability to maintain a high level of structural integrity under extreme conditions. These materials have excellent

resistance to fatigue and corrosion, which is crucial for the long-term storage of hydrogen. Additionally, composites are highly resistant to impact and thermal cycling, which is vital for safely transporting hydrogen. Another benefit of composite materials for hydrogen storage vessels is their ability to be tailored to specific requirements. Different matrix and reinforcement fiber combinations can achieve specific properties, such as high strength, modulus, or thermal stability. This allows engineers to optimize the design of the storage vessel for specific applications.

Composite materials are also highly cost-effective for hydrogen storage vessels. They are lightweight, which reduces the overall cost of the vessel, and they are also easy to fabricate, reducing manufacturing costs. Additionally, composite materials have a long service life and require minimal maintenance, which further reduces the overall cost of the vessel [68]. Some common materials used in the design of composite hydrogen storage vessels are presented in Figure 9.

| Carbon fibre reinforced plastic (CFRP) | •This composite material is made up of carbon fibres embedded in a plastic matrix. It is known for its high strength-to-weight ratio and is commonly used in the construction of hydrogen storage vessels. |
|---|---|
| Glass fibre reinforced plastic (GFRP) | •This composite material is made up of glass fibres embedded in a plastic matrix. It is known for its high strength-to-weight ratio and is often used in the construction of hydrogen storage vessels. |
| Aluminum metal matrix composite (AMMC) | •This composite material is made up of aluminum matrix reinforced with ceramic particles. It is known for its high strength and is often used in the construction of hydrogen storage vessels. |
| Titanium metal matrix composite (TMMC) | •This composite material is made up of titanium matrix reinforced with ceramic particles. It is known for its high strength and is often used in the construction of hydrogen storage vessels. |
| Stainless steel composite (SSC) | •This composite material is made up of stainless steel matrix reinforced with ceramic particles. It is known for its high strength and is often used in the construction of hydrogen storage vessels. |
| Magnesium metal matrix composite (MMC) | •Magnesium MMC is made by combining magnesium with a ceramic or metallic reinforcement. It is also lightweight and strong, but has less corrosion resistance than aluminum MMC. |

**Figure 9.** Composite materials for hydrogen storage.

This technology requires research and development to reduce costs while improving the performance, dependability, and durability of existing high-pressure vessels. Due to internal pressure, the composite shell can sustain significant mechanical loads. The extensive usage of carbon fiber accounts for 50–70% of the vessel's final cost (Figure 10). A composite structural optimization will allow for significant cost reductions in hydrogen devices. Numerical simulation needs to be improved because most engineers today work with simplified models that are frequently far from the real situation [69].

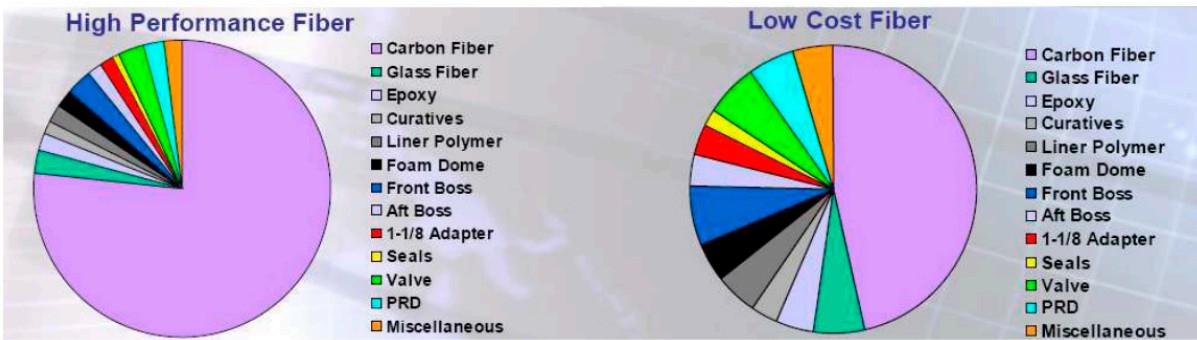

**Figure 10.** Costs of repartition of hydrogen high-pressure type IV vessels depending on carbon fiber types T1000/T700 (Quantum, USA) [69].

Carbon fiber type [70]: The type of carbon fiber used for hydrogen storage vessel fabrication can have an impact on the mechanical properties and hydrogen storage capacity of the vessel. Generally, high-strength carbon fibers such as T1000 or higher are preferred for hydrogen storage vessels because they offer superior mechanical properties, such as high tensile strength and stiffness, which are important for withstanding the high pressure and cyclic loading conditions that the vessels may be subjected to during hydrogen storage and transportation. However, the choice of carbon fiber type also depends on the specific application and requirements of the hydrogen storage vessel. For example, if the vessel needs to be lightweight, lower-strength carbon fibers such as T300 or T700 may be suitable. In addition, the manufacturing process and cost of the carbon fibers can also be a consideration.

Carbon fibers have many benefits, including high stiffness, high tensile strength, low weight, high chemical resistance, high-temperature tolerance, and low thermal expansion. We usually pay the most attention to the carbon content when purchasing carbon fiber products or parts. What type of carbon fiber material is used in manufacturing, on the other hand, is more important. Because different materials are used for different purposes, let us explain the differences between the T300, T700, and T800 carbon fiber. The different resins used in different carbon fiber materials are presented in Table 2.

**Table 2.** Type of carbon fiber.

| Type of Carbon Fiber | Sizing Type & Amount | Resin System Compatibility | Method |
|---|---|---|---|
| T300 | 40A/B (1.0%) | Epoxy | TY-030B-05 |
| | 40D (0.7%) | Epoxy | TY-030B-05 |
| | 50A/B (1.0%) | Epoxy, phenolic, polyester, vinyl ester | TY-030B-05 |
| T700S | 50C (1.0%) | Epoxy, phenolic, polyester, vinyl ester | TY-030B-05 |
| | 60E (0.3%) | Epoxy | TY-030B-05 |
| | F0E (0.7%) | Vinylester, compatible with epoxy | TY-030B-05 |
| T700G | 31E (0.5%) | Epoxy | TY-030B-05 |
| | 41E (0.5%) | Epoxy | TY-030B-05 |
| | 51C (1.0%) | Epoxy, phenolic, polyester, vinyl ester | TY-030B-05 |
| T800H | 40B (1.0%) | Epoxy | TY-030B-05 |
| | 50B (1.0) | Epoxy, phenolic, polyester, vinyl ester | TY-030B-05 |
| T800S | 10E (0.5%) | Epoxy | TY-030B-05 |
| | 50C (1.0%) | Epoxy, phenolic, polyester, vinyl ester | TY-030B-05 |

T300 is the first carbon fiber. It was made by the Japanese company TORAY in the 1970s. It is a standard modulus carbon fiber with a fiber modulus of 33–34 Msi or slightly higher. So, the T300 standard modulus carbon fibers made by Toray are a standard in the industry. T700 is a standard carbon fiber with a high tensile strength and a standard modulus. It is also a standard industrial fiber. From a performance standpoint, carbon fibers T300 and T700 both have a tensile modulus of 230 GPa and a diameter of 7 m. However, T300 has a tensile strength of 3.53 GPa and T700 has a tensile strength of 4.90 GPa. This means that T300's strength increased by 38.8%, T700's elongation increased by 40%, and T700's volume density increased by 2.27%. T800 is an intermediate modulus with a high tensile strength fiber and a tensile modulus of 42 Msi. It has high-level and balanced composite properties.

Because it was the first composite material, T300 carbon fiber has been used a lot for 30 years. At the start, its price was very high. The price of a $400 \times 500 \times 5$ mm carbon fiber sheet of T300 material was between USD 130 and USD 140. However, Toray Corporation began shipping more and more T300, so the cost of the material began to go down. The price of T300 carbon fiber is now very low. A 4,005,005 mm carbon fiber plate costs about USD 80 now, which is almost half of what it cost when it first came out. In terms of price, T700 is about 40% more expensive than T300. A $400 \times 500 \times 5$ mm T700 carbon fiber sheet costs about USD 120, while a T800 sheet costs between 30% and 40% more than a T700 sheet. The T700 grade and the cost of raw materials, and the cost of making and processing T800 grade carbon fiber, are different from those of T300. Because of this, the final price of T700 and T800 is much higher than that of T300.

T300 is the most common and least expensive type of carbon fiber. Drone fans like T700 because it is a good material. It is used a lot in the drone industry because it is strong and has good value for money. The most expensive is T800 carbon fiber, which is also the strongest. It is mostly used to make high-end products. To make the product lighter as a whole, T800 carbon fiber material must also be used.

Several key considerations must be considered when designing a composite hydrogen storage vessel. Two of the most important considerations are the pressure and temperature requirements of the vessel. Hydrogen storage composite vessels must withstand extremely high pressures, typically 350 to 700 bar, and low temperatures of 20 K without failure or deformation. Another critical consideration is the composite materials' strength-to-weight ratio [71]. Hydrogen storage vessels must be lightweight and highly durable, and composite materials are an excellent option as they have a high strength-to-weight ratio. Additionally, the reinforcement fibers used in the composite materials must be chosen carefully to ensure they can withstand the high pressures and temperatures required. The vessel's design must also consider the safety and security of the hydrogen storage. The vessel must be designed to prevent leaks or spills and withstand impact and thermal cycling, which is particularly important for safely transporting hydrogen. Designing a composite hydrogen storage vessel requires careful consideration of the materials and structural design to ensure safety and durability. Some key factors are presented in Figure 11.

Recently, researchers [72] have investigated the impact of polyethylene glycol (PEG) modification on the properties of the composite layer used in cryo-compressed hydrogen storage vessels (HSVs) (Figure 12). The results show that the modified epoxy resin retains its high strength of 73.38 MPa while decreasing its elastic modulus by 45.31%. The 3 wt% PEG 600 modification reduces winding plies by 32.65% while increasing the carbon fiber strength usage rate by 50.69%. Carbon fiber mechanical qualities can be used in $CcH_2$ storage vessel design with only a 6.06% increase in winding plies while maintaining improved safety.

The application of new materials such as graphene and carbon nanotubes (CNTs) is becoming increasingly important for the fabrication of hydrogen storage vessels. These materials have several advantages, including a high surface area for hydrogen adsorption, high strength and stiffness for improved mechanical properties, lightweightedness for reduced weight, flexibility for complex geometries, and scalability for cost-effective production. These advantages can help improve the performance, safety, and cost-effectiveness of hydro-

gen storage technologies, which are crucial for a sustainable energy future. However, more research is needed to fully understand the properties and behavior of these materials in hydrogen storage applications, as well as their long-term durability and cost-effectiveness.

**1** • Wall thickness: The thickness of the walls of the vessel will depend on the pressure and temperature requirements of the hydrogen stored within.

**2** • Joints and connections: Careful consideration must be given to the design of joints and connections to ensure that the vessel can withstand high pressures and temperatures without leakage or failure.

**3** • Heat dissipation: Hydrogen storage vessels must be designed to dissipate heat effectively to prevent overheating and possible explosion.

**4** • Impact resistance: The vessel must be able to withstand impacts and collisions without damage or failure.

**5** • Corrosion resistance: The materials used in the design of the vessel must be resistant to corrosion to ensure long-term durability.

**Figure 11.** Critical considerations for the technical design of hydrogen storage.

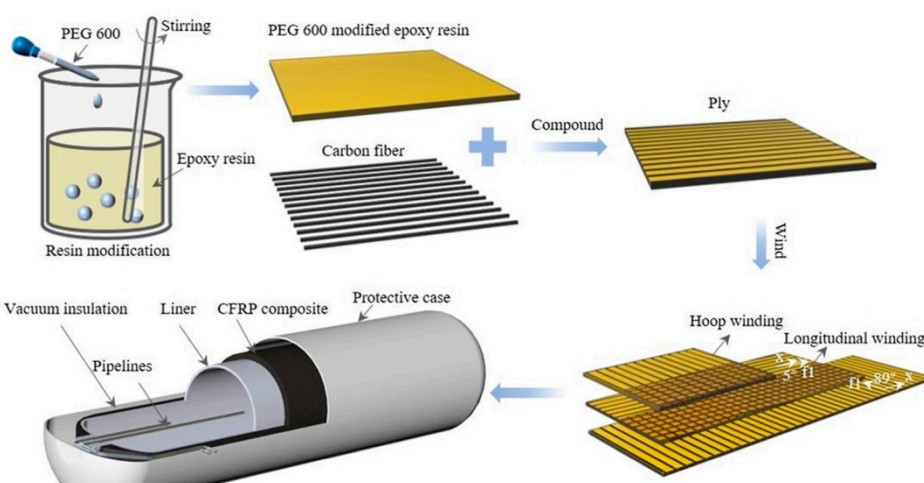

**Figure 12.** Properties improvement of a composite layer of a cryo-compressed hydrogen storage vessel [72].

Other projects have aimed to develop high-performance and lightweight Graphene-CFRP compressed hydrogen storage tanks that are becoming increasingly popular for aerospace applications [73]. These tanks are designed to store and transport hydrogen safely and efficiently while also being lightweight and durable enough for aerospace applications. One key component of these tanks is the nanomaterial-reinforced 3D printed polymer liner, which provides a barrier to hydrogen permeation and prevents hydrogen from escaping. Another critical component is the graphene low permeability layer, which provides additional protection against hydrogen permeation. Finally, the graphene and related materials-reinforced matrix CFRP composite overwrapped layer provides a

strong, durable, and lightweight outer layer that can withstand the harsh conditions of aerospace applications (see Figure 13). These advanced materials and design elements make Graphene-CFRP compressed hydrogen storage tanks ideal for use in aerospace applications and help promote clean energy sources.

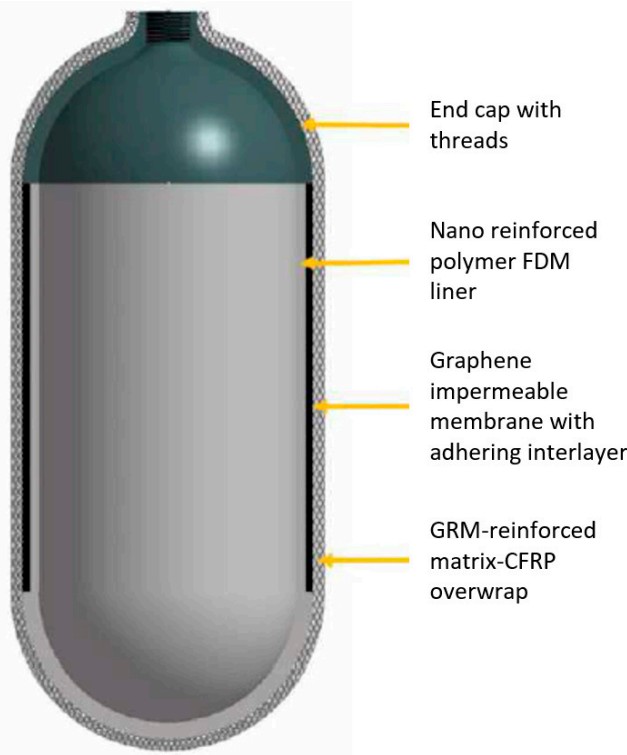

**Figure 13.** High-performance and lightweight Graphene-CFRP compressed hydrogen storage tanks [73].

According to carbon nanotube producer OCSiAl [74], using carbon nanotubes (CNTs) in hydrogen storage tanks offers several advantages over traditional materials. CNTs are known for their high strength-to-weight ratio, high thermal and electrical conductivity, and excellent resistance to hydrogen permeation. This makes CNTs ideal for hydrogen storage tanks, as they can provide a strong and lightweight container to prevent hydrogen from escaping. OCSiAl also claims that CNTs can be used to create more efficient hydrogen storage tanks, as they can store hydrogen at higher pressures and temperatures than traditional materials. This means that more hydrogen can be stored in a smaller volume, reducing the size and weight of the storage tank. Additionally, CNTs can be easily integrated with other materials, such as composites, to create hybrid hydrogen storage tanks with improved performance and durability (Figure 14).

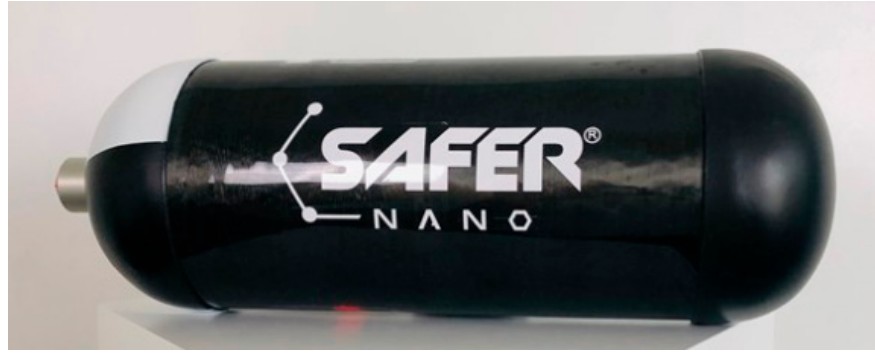

**Figure 14.** CNT-enhanced composite tanks [74].

The same researchers from OCSiAl [74] showed that the graphene nanotubes introduced in composite reinforcement bring improvements in interlaminar shear strength (ILSS), leading to an increase in burst pressure level by up to 30%, according to the results of cylinder impact tests. This incredible improvement in impact resistance has allowed us to reduce the weight of the cylinder while maintaining its mechanical properties, which resulted in the lightest 6.8-L cylinder in the world for 300 bar of working pressure. The total mass of our SAFERnano cylinder, including all protective attachments and coatings, is less than 2.8 kg. This new generation of cylinders achieved a weight reduction of up to 75% compared with competing solutions, and of 15% compared with our previous generation of products.

The recycling of composite hydrogen storage vessels is an important issue as these vessels have the potential to be an environmentally friendly alternative to conventional fuel storage methods. However, composite materials used in hydrogen storage vessels can be challenging to recycle due to their complex composition and the presence of hydrogen in the vessels. There have been several recent studies on the recycling of composite hydrogen storage vessels, including the use of mechanical and thermal treatments to separate the various components of the composite material. For example, the CETIM group [75] has explored mechanical grinding and shredding methods to reduce the composite material to small pieces, which can be further processed to recover the fibers and resin (Figure 15). Other studies have investigated using thermal treatments, such as pyrolysis, to break down the composite material into its constituent parts. In addition to these physical recycling methods, there have also been studies on reusing composite hydrogen storage vessels by repurposing the vessels for other applications or refurbishing and refilling the vessels with hydrogen. However, these methods require careful consideration of the integrity and safety of the vessels to ensure that they remain safe for use.

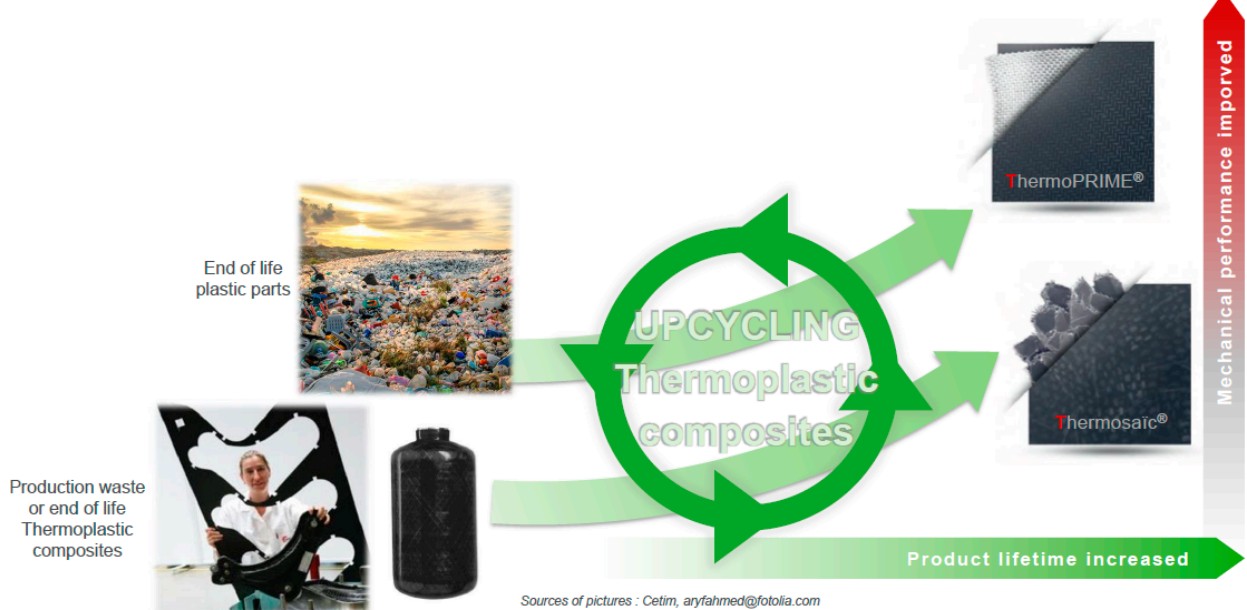

**Figure 15.** Thermosaïc® Technology to recycle end-of-life tanks [75].

## 5. Finite Element Analysis

The analytical design solution for composite hydrogen tanks is based on broad assumptions about load and boundary conditions and does not consider changes in stiffness near the polar boss. To accurately predict how filament wound pressure vessels will behave, finite element analysis (FEA) must be used to model these and other effects correctly. Most filament-wound pressure vessels have first-order non-linear geometry effects that the FEA can only capture. The optimization seeks to increase the composite vessel's weight, strength,

dependability, and service life by designing the wound angles and thickness. As design methods and production technique advance, the composite vessel adopts a lightweight development strategy. Weight reduction has become a method bottleneck that restricts the composite vessel's economy and practicability. Building a material-structure-processing integrated design and computation methods for composite vessels is imperative in this situation. Other manufacturing and testing factors can affect a composite vessel's performance; hence the design's importance may need to be examined. Nonetheless, the significance of numerical simulation and artificial intelligence is anticipated to be further emphasized by this review.

The finite element analysis has evolved into a potent tool for solving numerical issues associated with complicated structures. Using a combination of continuum damage mechanics (CDM) and FEA, the progressive failure analysis of the composite vessel accounts for the stiffness degradation and predicts the rupture pressure [76]. In particular, the FE approach that implements the damage model has attracted considerable scholarly interest. Several intelligent algorithms that imitate the evolutionary concepts of computational biology, physics, and immunology have been utilized to optimize composite vessels. Genetic algorithms, simulated annealing, and artificial immune system are standard algorithms. Multiple commercial algorithms exist today for generating filament winding pathways, and they all rely heavily on the shape of the mandrel. In order to enhance the final product's performance, researchers have investigated a more complicated path-generating approach, as detailed in the work presented in [77]. The approach considered the mandrel's evolving shape due to the ply's unequal thickness distribution from previous winding processes. The alternative strategy involves using high-tech machinery and paying for commercial software for path creation to arrive at an optimal design for the COPV. Using a simulation tool like WoundSim [42] can help reduce the time and cost of developing composite hydrogen storage tanks, allowing designers to test and optimize their designs without needing physical prototypes (Figure 16). This can be particularly useful when working with composite materials, as these materials can be difficult to work with and require specialized manufacturing processes. Advanced features like optimization, parametric design, and creation of experiments are available in WoundSIM. The translation efficiency in FEA is defined as the ratio of the observed failure strain to the theoretical composite tensile strain. The model's fiber quality, winding method, manufacturing process, and actual structure vary (e.g., voids, fiber misalignment, resin pockets, etc.). Each tank maker has varied translation efficiency, discovered via experience. The FEA model must be calibrated to obtain translation efficiency and account for the abovementioned changes. WoundSim is a set of comprehensive features that ensure a rapid design and simulation of pressure vessels. Overall, using simulation tools like WoundSim is an important part of the hydrogen storage vessels industry, allowing designers to create more efficient and effective products using advanced materials. The main features are listed below:

- A democratized tool with a comprehensive and standalone user interface.
- User interface short time response, which accelerates design duration.
- Comprehensive and well-chosen design parameters allow a quick variation of the layers' shape.
- Fully automated FEA model generation, allowing performing simulations with minimum FE knowledge.
- Winding problems anticipation.
- Smart layup rendering allows layer selection and intersection detection.
- Design of experiment capabilities for parametric design optimization.
- Full compatibility with ABAQUS software and no need for a FORTRAN compiler to post-process specific material outputs.
- Compatibility with filament winding software.
- Models calibration and correlation with produced reservoir measurements.

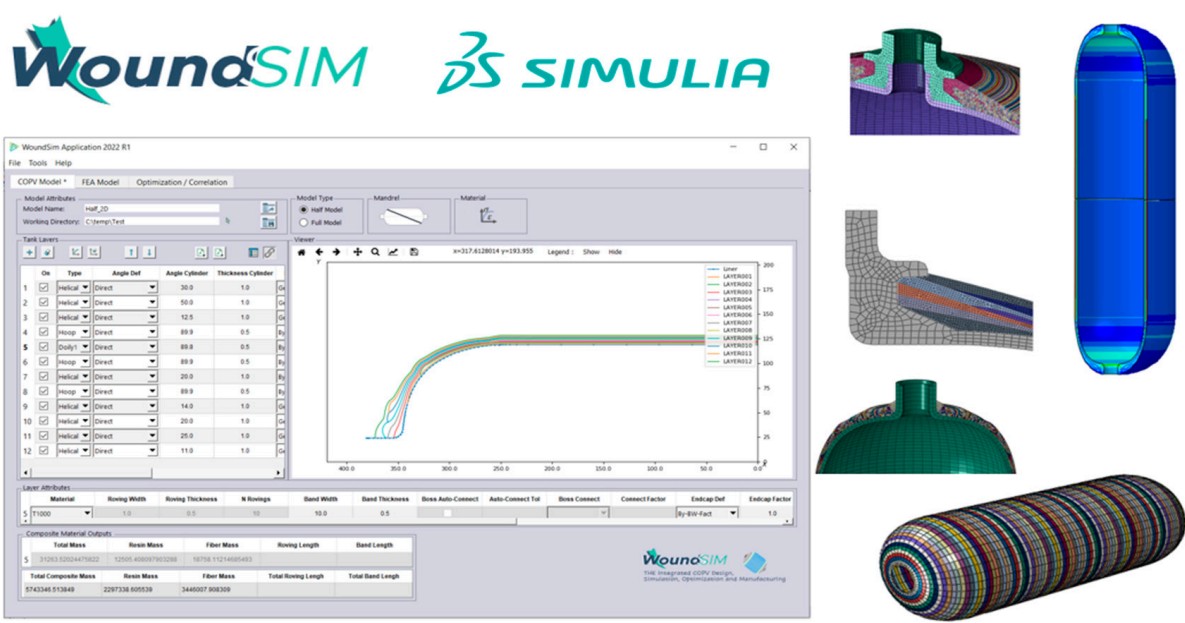

**Figure 16.** WoundSIM software [42].

Commonly, FEA is used to investigate the behavior of the directional mechanical properties of carbon fiber composite material, which is particularly important for hydrogen storage pressure vessels (Figure 16). When addressing hydrogen pressure vessels subject to internal loads, many failure modes, including fiber deterioration and matrix damage, are frequently examined [78]. Leh et al. [79] published a well-developed progressive damage model to explore the failure behavior of a Type IV hydrogen storage pressure vessel; the work presented two FE models to mimic situation 28, in which the vessel's internal pressure builds until the composite vessel fails abruptly. Nguyen et al. [80] completed considerable research on damage modeling of various composites and developed an outstanding instrument for analyzing the damage status of hydrogen storage vessels subjected to thermomechanical loads.

Bogenfeld et al. [81] investigated the performance of composite materials following impact using both analytical and experimental methods, providing an overview of the design of damage-tolerant composite structures. Wang et al. [82] used material property degradation and continuum damage mechanics to determine a composite vessel's ultimate strength and complex failure behaviors; a totally overwrapped FE model was used to examine multiple failure modes. Liu and his colleagues [27] provided a comprehensive review of research related to the design and optimization of composite high-pressure hydrogen storage vessels. The authors focused on numerical simulation techniques used to model the behavior of composite materials under high-pressure hydrogen storage conditions. These techniques include FEA and computational fluid dynamics (CFD) simulations. This review also examined recent studies that aimed to optimize the design of composite hydrogen storage vessels by exploring different design parameters, such as the thickness and orientation of the composite fibers, the type of resin used, and the overall shape of the vessel. The authors concluded by highlighting future research directions in the field, including the need for more advanced simulation tools, the development of new composite materials, and the exploration of alternative hydrogen storage methods. In another research study [83], the authors reviewed existing research on the composite shells used in high-pressure hydrogen storage vessels. They also summarized simulation techniques such as FEA and CFD simulations to model the behavior of composite shells. Additionally, the authors covered studies that optimized the design of composite shells by examining parameters like fiber thickness, resin type, and vessel shape. Ramirez et al. [84] presented convincing results from a simulation of a Type IV pressure vessel's safe/unsafe burst mode. Zaami et al. [85]

performed a comprehensive numerical analysis of the heat flow and temperature distribution around the nip point during the helical winding of fiber-reinforced thermoplastic tapes on a cylinder-shaped mandrel. This article focused primarily on the relevant stress and strain response to specific internal stressors. In conclusion, FE analysis can be utilized to design hydrogen storage tanks with acceptable accuracy and cost-effectiveness, but researchers must use surrogate models on extremely large systems to save computing time.

### 5.1. Multi-Scale Modeling of Composite Hydrogen Storage Vessels

Multi-scale modeling of composite hydrogen storage vessels involves using mathematical models and simulation tools to analyze the performance of these vessels at different scales (Figure 17). This analysis can include modeling the behavior of individual materials and components at the microscale and predicting the vessel's overall performance at the macroscale [86–89]. Multi-scale modeling is useful for understanding the complex interactions within composite hydrogen storage vessels and identifying the factors influencing their performance. It can also be used to optimize the design and construction of these vessels by identifying the materials and techniques that result in the best performance. To perform multi-scale modeling of composite hydrogen storage vessels, researchers may use software such as ABAQUS or ANSYS (Figure 18) and specialized modeling and simulation tools designed for this purpose [88]. They may also use experimental data from tests and simulations to validate and refine their models. Overall, the goal of multi-scale modeling is to develop a better understanding of the behavior of composite hydrogen storage vessels and improve their performance.

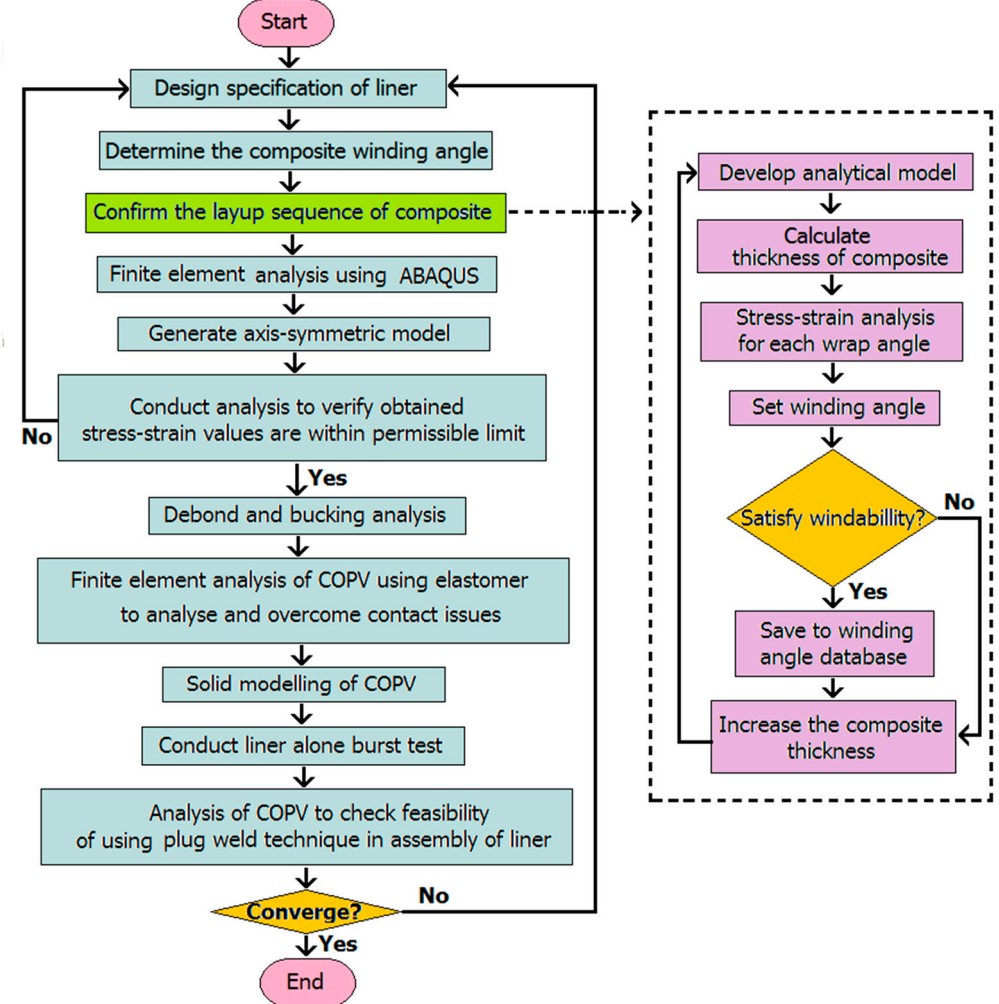

**Figure 17.** Flow chart of finite element modeling of a composite vessel [82].

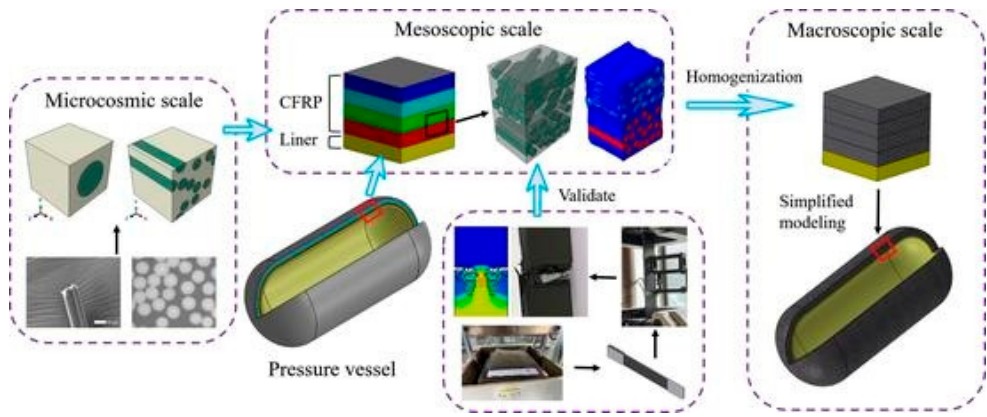

**Figure 18.** Multiscale study of CFRP pressure vessel [88].

In recent years, some new failure criteria have driven the development of failure analysis, providing information on the interaction between the microscopic and macroscopic aspects of composite materials for failure analysis. The post-initial failure behavior of the composite laminate structure is simulated by the material property degradation method, especially the combination of CDM and the commercial finite element analysis method. The multi-scale failure analysis was progressively developed by new finite element methods, such as the cohesive element (CE) and the representative volume elements (RVE) methods. The following summarises the progressive failure analysis of composite tanks using the finite element method as a reference for subsequent failure analysis [38]. The flow chart of the progressive failure analysis is shown in Figure 19, which generally consists of (I) stress analysis, (II) failure evaluation, (III) material degradation, and (IV) burst pressure detection.

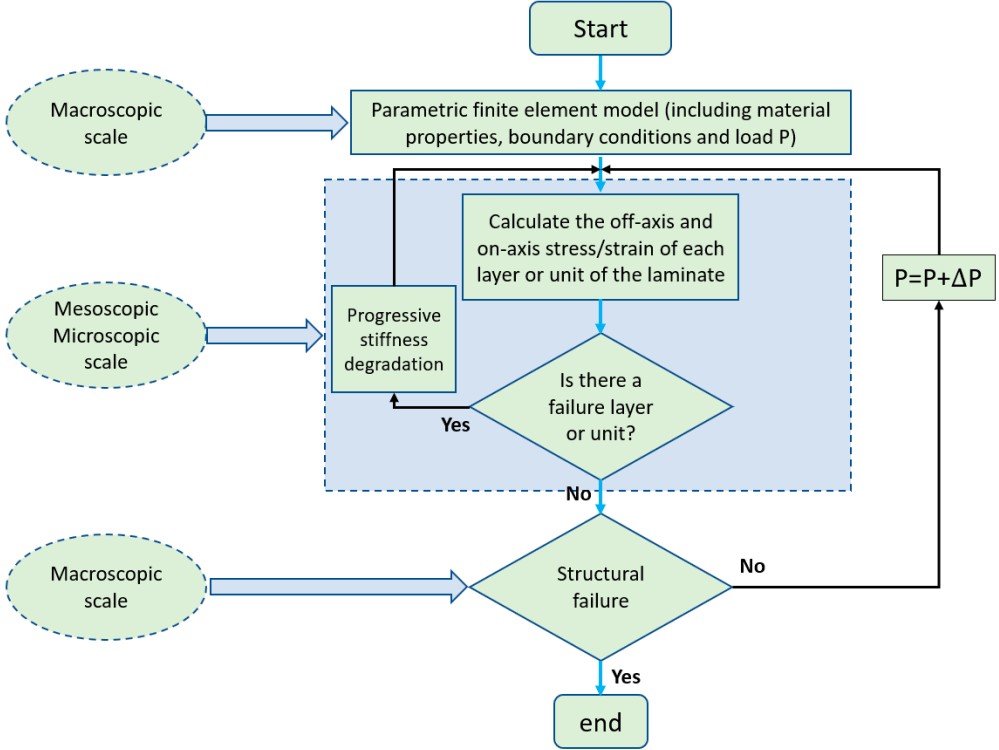

**Figure 19.** The flow chart of the progressive failure analysis [38].

Recently, Nguyena and his colleagues [90,91] developed multi-scale models for analyzing pressure vessels. These models explored the meso-macro or micro-meso-macro approaches to computing the macroscopic vessel response. Models using the meso-macro

approach compute the vessel's behavior from the composite lamina given a priori. In contrast, models exploring the micro-meso-macro approach predict the composite vessel behavior from those of the microscale constituents (i.e., fiber and matrix). In addition to these approaches, micro-macro approaches use a reference volume element or unit cell to obtain the damage state at the constituent level to be fed into the macroscopic FE model of a pressure vessel.

Also, Lin et al. [92] presented a new approach to modeling and analyzing the progressive failure of composite pressure vessels. The study aimed to improve the accuracy and efficiency of the design process for composite pressure vessels, which are commonly used in various applications, including hydrogen storage. The authors used a multiscale modeling approach that combined micromechanical analysis, mesomodeling, and macromodeling to simulate the behavior of the composite pressure vessels under different loading conditions (Figure 20). They also incorporated the Puck failure criterion to predict the onset of damage and the propagation of cracks in the composite material.

Nevertheless, many of the modeling works on pressure vessels cited above applied the meso-macro approach in combination with the classical lamination theory (CLT) or a finite element (FE) method associated with a failure criterion, such as Hashin, Puck, Tsai-Hill, and Tsai Wu. Using a failure criterion requires lamina's strength data obtained through mechanical testing on flat specimens, which could represent a tremendous amount of effort given the temperature-dependent mechanical property data needed. In addition, as pointed out in [93], the micro-meso-macro approach offers a more efficient way than the meso-macro approach for obtaining the lamina behavior in complex laminated composite structures, such as filament-wound pressure vessels, for which data obtained from conventional flat specimens do not necessarily reflect actual mechanical properties of the layers in the vessels. While continuum damage mechanics (CDM) has been significantly explored to model progressive damage in laminated composite structures [94], its applications to pressure vessels have still been limited. Transverse matrix cracking that obeys a damage evolution relation within the continuum damage mechanics (CDM) framework can progressively evolve from initiation to saturation state [95–98]. Composite failure by fiber rupture during vessel loading is predicted by a micromechanical fiber failure criterion that accounts for fiber strength and matrix stress [99]. A novel recursive multi-scale modeling to predict the burst pressure in filament wound composite pressure vessels was developed by Rafiee [100]. The modeling covered all scales of micro, meso and macro, and the unavoidable imperfections associated with fiber arrangement during the filament winding process were considered (Figure 21).

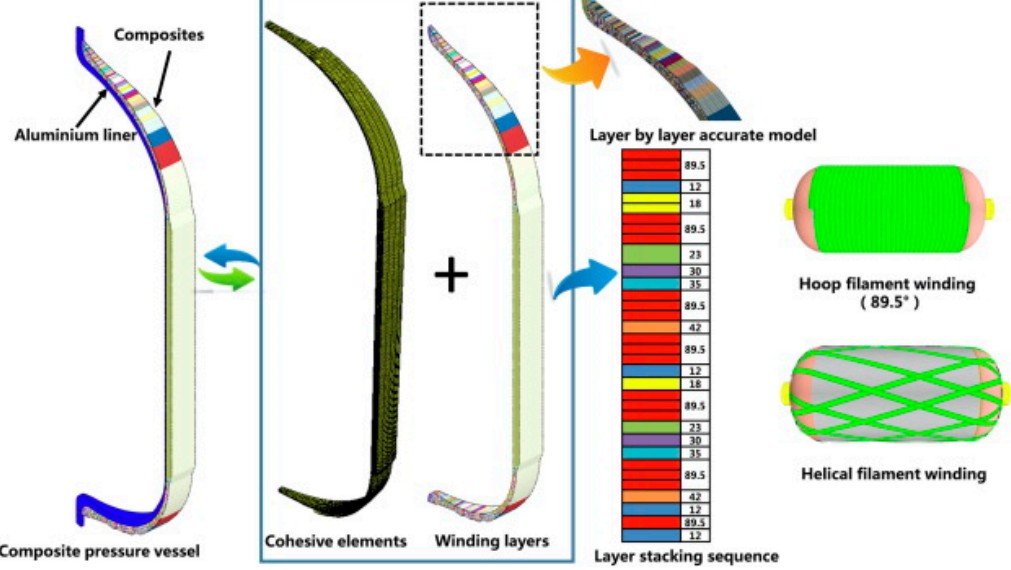

**Figure 20.** Progressive damage analysis for the multiscale modeling of composite pressure vessels [92].

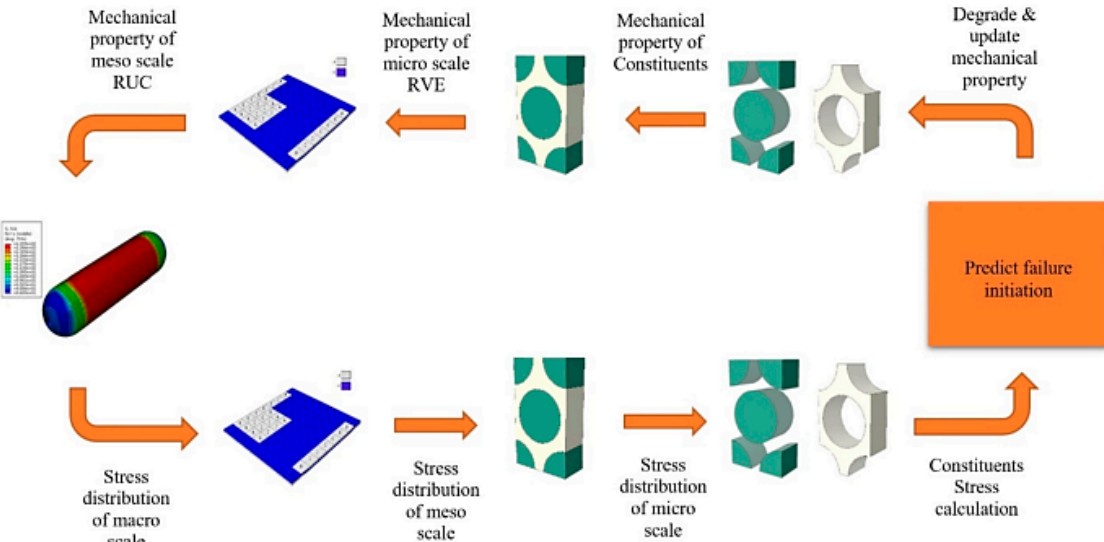

**Figure 21.** Multi-scale modeling to predict the burst pressure in filament wound composite pressure vessels [94].

The development of test procedures for hydrogen storage vessels is critical for ensuring the safety and performance of these vessels. Several tests can be conducted to evaluate the structural integrity and performance of hydrogen storage vessels, including impact and static fatigue tests. Impact tests are designed to evaluate the ability of a hydrogen storage vessel to withstand sudden impacts or loads. These tests involve subjecting the vessel to a sudden impact or load, such as a drop or impact from a falling object, and measuring the vessel's response. The results of these tests can be used to assess the vessel's resistance to impact damage and to identify any potential weak points in the vessel's design. In addition, static fatigue tests are designed to evaluate the ability of a hydrogen storage vessel to withstand repeated loading over time. These tests involve subjecting the vessel to repeated loads, such as pressurization and depressurization cycles, and measuring the vessel's response. The results of these tests can be used to assess the vessel's resistance to fatigue failure and identify any potential fatigue cracking or other damage. To develop test procedures for hydrogen storage vessels, it is crucial to consider the vessel's specific design and intended use [101–104]. The test procedures should be tailored to the vessel's specific requirements and operating conditions and performed according to relevant industry standards and guidelines (Figures 22 and 23).

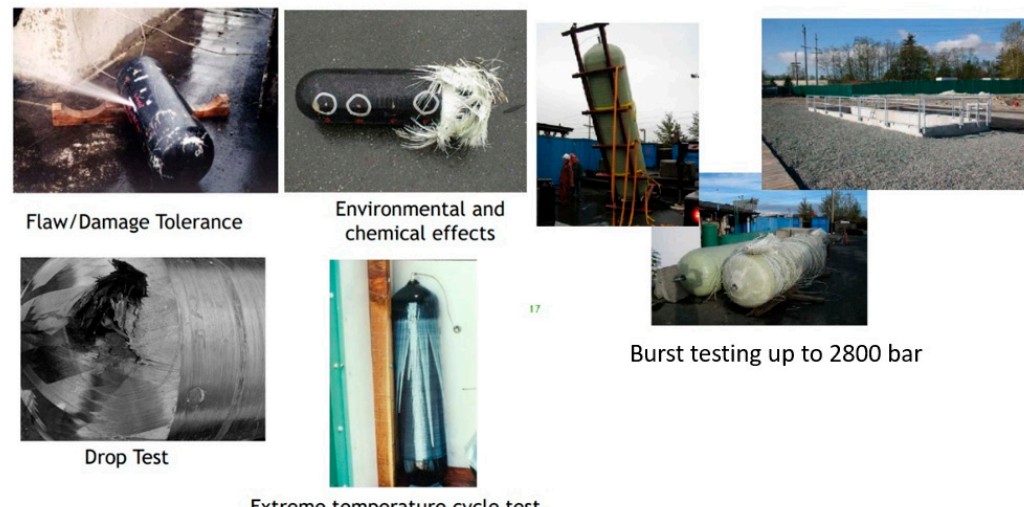

**Figure 22.** Hydrogen tank safety testing [105].

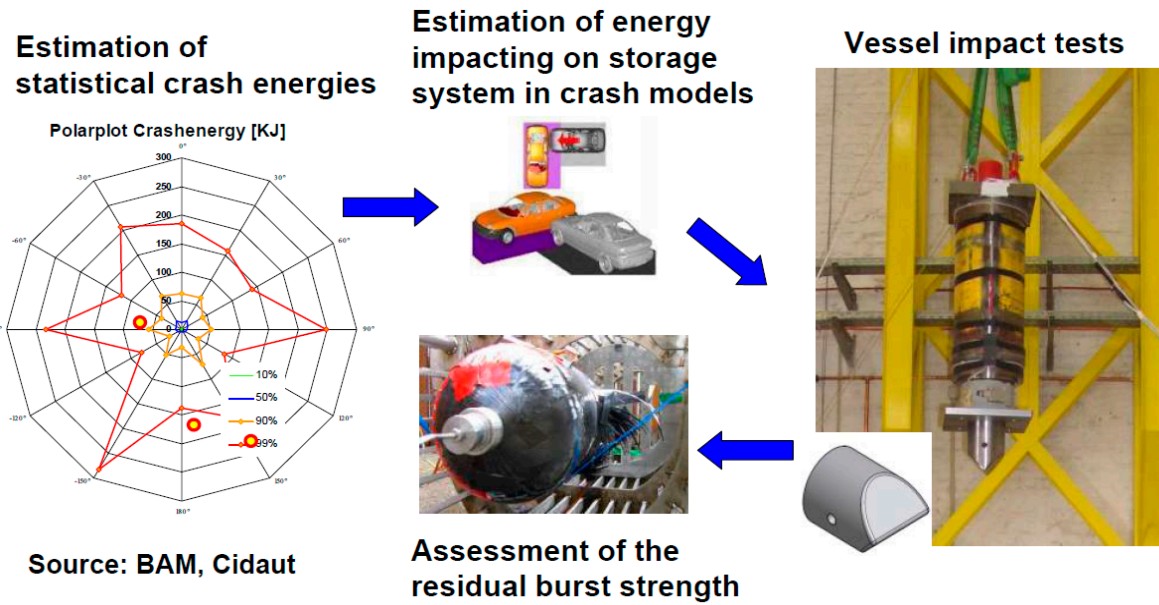

**Figure 23.** The procedures tailored to the specific requirements and operating conditions of the vessel [106].

*5.2. Development of Test Procedures for C-H$_2$ Vessels: Impact Test/Static Fatigue*

Hydrogen permeability in carbon fiber composite structures is important when designing hydrogen storage vessels. Hydrogen permeability refers to the ability of hydrogen atoms to pass through a material and is a measure of the material's gas transmission rate [107]. The permeability of hydrogen in carbon fiber composite structures can vary depending on the specific composite material used, the thickness of the material, and the temperature and pressure conditions. Carbon fiber composite materials generally have low permeability to hydrogen, making them suitable for use in hydrogen storage vessels. The permeability of hydrogen in the inner and outer vessels of a composite hydrogen storage vessel can also vary depending on the specific materials and design of these vessels. The inner vessel, typically designed to hold the hydrogen gas, may have a higher permeability than the outer vessel, which is designed to provide structural support and protect the inner vessel from external factors such as impact and temperature fluctuations (Figure 24).

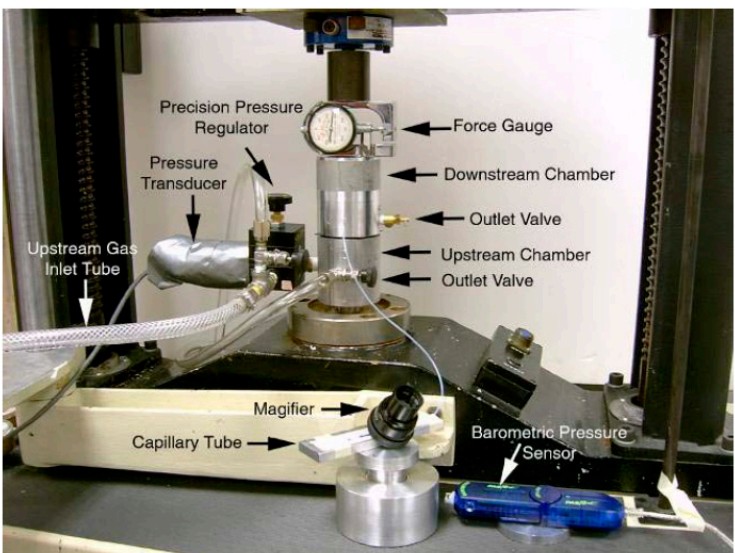

**Figure 24.** Permeability Test Facility [108].

A series of permeability tests can be performed to determine the hydrogen permeability in a carbon fiber composite structure. These tests involve exposing the material to a known concentration of hydrogen gas and measuring the amount of hydrogen that passes through the material over a specified period. The results of these tests can be used to determine the material's gas transmission rate and to optimize the design of the hydrogen storage vessel (Figure 25).

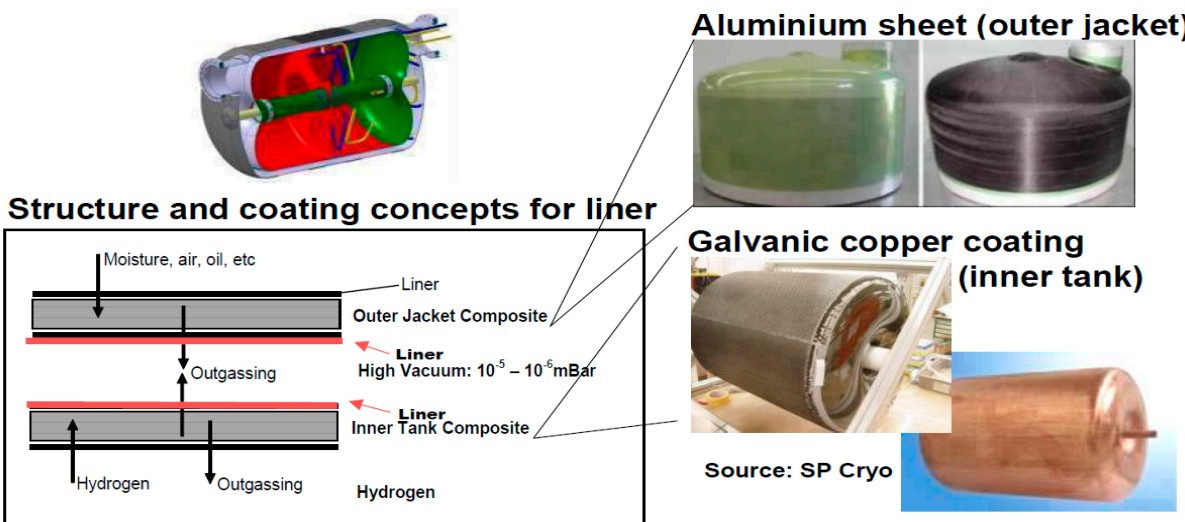

**Figure 25.** Development of test procedures for C-H$_2$ vessels [106].

Overall, the research on hydrogen permeability in composite hydrogen storage vessels has contributed significantly to developing safer and more efficient hydrogen storage technologies. These efforts have helped improve the performance and reliability of composite hydrogen storage vessels and support hydrogen growth as a clean and renewable energy source. Several types of complex loading can affect the performance and safety of composite hydrogen storage vessels (Figure 26). These loading conditions can include the following:

- Internal pressure: The internal pressure of a hydrogen storage vessel can vary due to the filling and emptying of the vessel, as well as changes in temperature and ambient pressure.
- Thermal loading: The temperature of a hydrogen storage vessel can vary due to the ambient temperature, heat generated during the filling and emptying, and the exothermic reactions of hydrogen.
- Cyclic loading: The filling and emptying of a hydrogen storage vessel can result in cyclic loading, which can cause fatigue damage and reduce the vessel's lifespan.
- Impact loading: Sudden impacts or loads, such as a drop or impact from a falling object, can cause damage to a hydrogen storage vessel.
- External loads: External loads, such as the vessel's weight and any additional equipment or materials, can affect the structural performance of a hydrogen storage vessel.
- Corrosion: The presence of moisture or other corrosive agents can cause corrosion and weaken the structural integrity of a hydrogen storage vessel.
- Vibration: Vibration can cause fatigue damage and reduce the lifespan of a hydrogen storage vessel.
- Thermal expansion: Temperature changes can cause the composite material of a hydrogen storage vessel to expand or contract, affecting the vessel's structural performance.

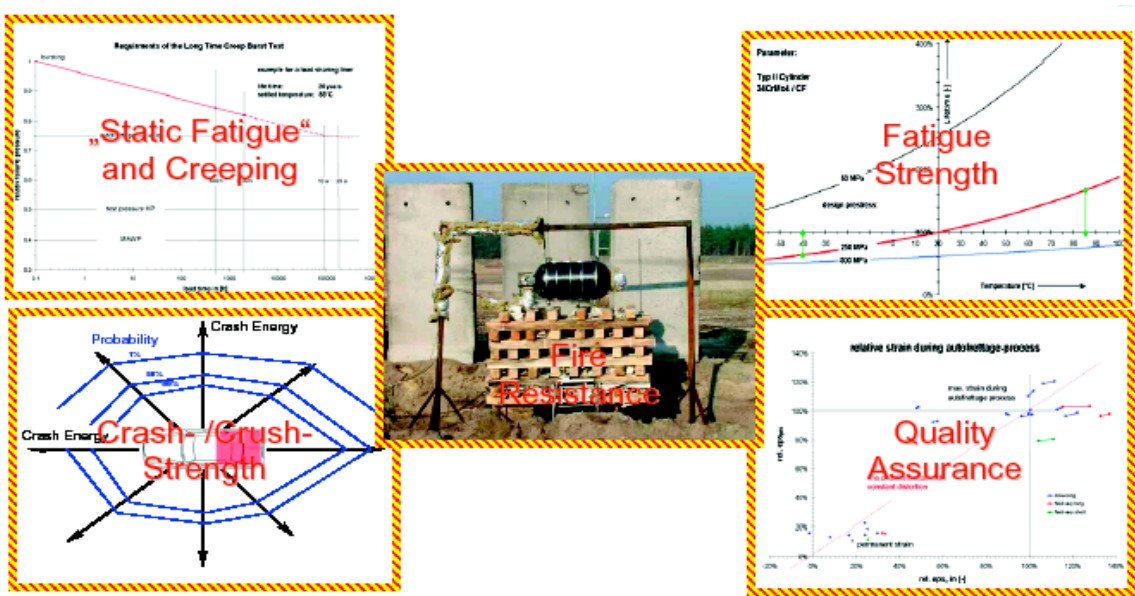

**Figure 26.** Carbon fiber composite structure of the inner tank and outer jacket [106].

## 6. Artificial Intelligence and Optimization Models for Hydrogen Storage Vessels Design

Artificial intelligence (AI) is increasingly being used to improve the design and performance of hydrogen storage vessels [109]. AI can analyze large amounts of data and identify patterns and relationships that can help optimize the design of the vessels and improve their safety and performance. In addition, the use of AI in the design and optimization of composite storage vessels has the potential to improve these systems' performance, safety, and sustainability. By automating many of the manual and time-consuming tasks associated with design and analysis, AI can help accelerate the development of effective and efficient composite storage solutions. Several artificial intelligence models can be used to design hydrogen storage vessels (Figure 27).

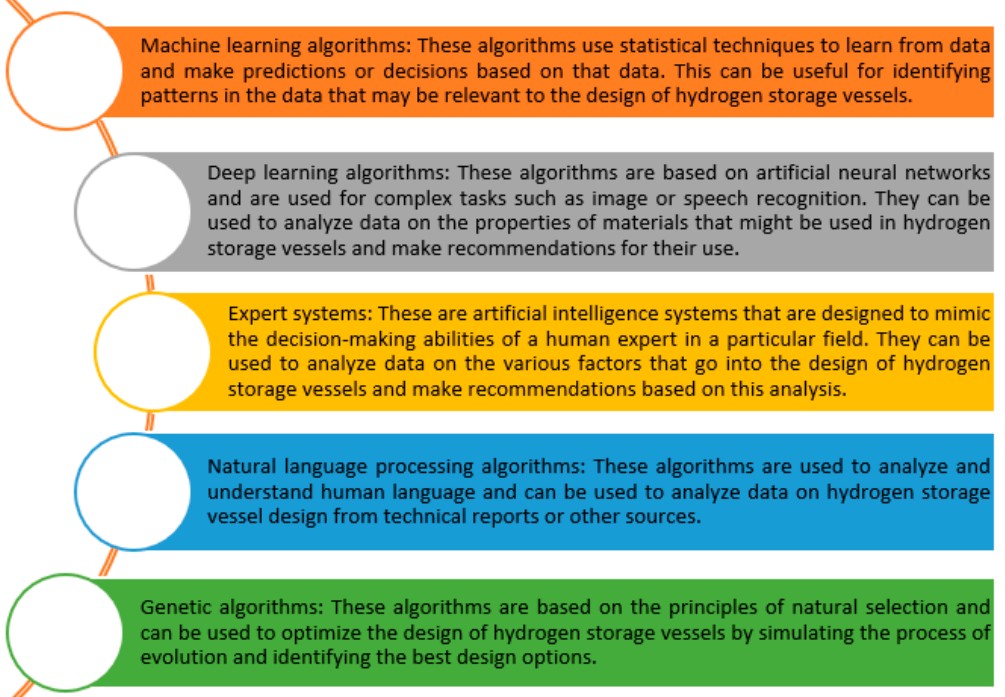

**Figure 27.** Types of artificial intelligence models used for the design of hydrogen storage vessels.

Machine learning models can be trained on data from experiments or simulations to identify materials that have desirable properties, such as high hydrogen storage capacity, low weight, or high strength. By using machine learning to guide the selection of materials, researchers can more quickly and efficiently identify promising candidates for use in hydrogen storage systems (Figure 28). Another area of research involves the use of reinforcement learning to optimize the design of hydrogen storage vessels. Reinforcement learning involves training a machine learning model to learn from feedback received based on its actions, with the goal of maximizing a specific reward function. In the context of hydrogen storage vessel design, this could involve optimizing the geometry, material properties, or other design parameters of the vessel to maximize hydrogen storage capacity or minimize weight. The source of the training set for hydrogen storage vessel design will depend on the specific problem statement and available data. In general, the training set can be sourced from experimental or simulation data, or from existing datasets in the literature or industry. If experimental data is available, it can be used to train a machine learning model to predict the behavior of hydrogen storage vessels under different conditions. This can involve testing the vessels under a range of pressures, temperatures, and other relevant variables, and collecting data on their performance. Alternatively, simulation data can be generated using software tools that simulate the behavior of hydrogen storage vessels based on input variables, such as material properties, geometrical features, and operating conditions. Once the training set is selected, the testing set can be chosen using a range of techniques. One common approach is to randomly partition the dataset into a training set and a testing set, with a majority of the data used for training and a smaller subset used for testing the accuracy and generalization of the model. Another approach is to use cross-validation, where the dataset is divided into several folds and the model is trained and tested on each fold in turn. It is important to ensure that the testing set is representative of the data that the model is expected to encounter in real-world scenarios. This means that the testing set should include a diverse range of data points, and should not contain any data points that were used in the training set. By carefully selecting and partitioning the training and testing sets, we can ensure that the machine learning model is accurate and reliable for predicting the behavior of hydrogen storage vessels.

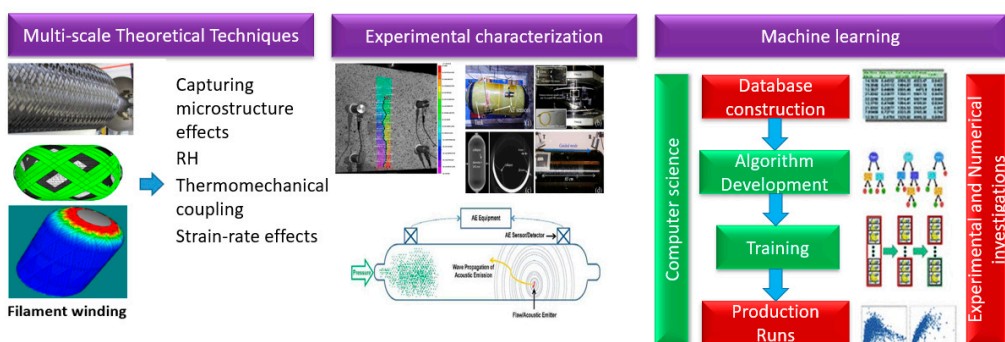

**Figure 28.** Machine learning fast prediction using numerical and experimental results.

An objective function is required to optimize a composite pressure vessel [110]. This function must represent the performance of the composite material, and in some instances, local minima may exist. This objective function is utilized by optimization techniques to discover the optimal solution. In general, these methods can be divided into two categories: those that require simple function evaluations and those that require the computation of gradients. Due to the complexity of the mechanical behavior of composite materials, gradient calculation may be tricky. Gradient-based approaches are, therefore, ineffectual when several design variables are present and may result in a local minimum, rendering them untrustworthy. Therefore, metaheuristics such as genetic algorithms (GA) or simulated annealing (SA) may be more effective for optimizing composite pressure vessels. Even though metaheuristics cannot guarantee a globally optimal solution, they can frequently produce

better outcomes with less computational work than algorithms, iterative approaches, or basic heuristics by searching a broad collection of potential solutions.

Islam et al. (2018) [111] conducted an experimental investigation using feature selection and deep learning techniques to classify cracks in a pressure vessel. Acoustic emission signals were used to detect the presence of cracks in the pressure vessel. Using AE data obtained from a self-designed pressure vessel, the efficacy of the proposed method (GA + DNN) was illustrated. The experimental findings demonstrated that the proposed method was highly effective for selecting discriminant features. These features were used as input to the DNN classifier, achieving a classification accuracy of 94.67% (Figure 29).

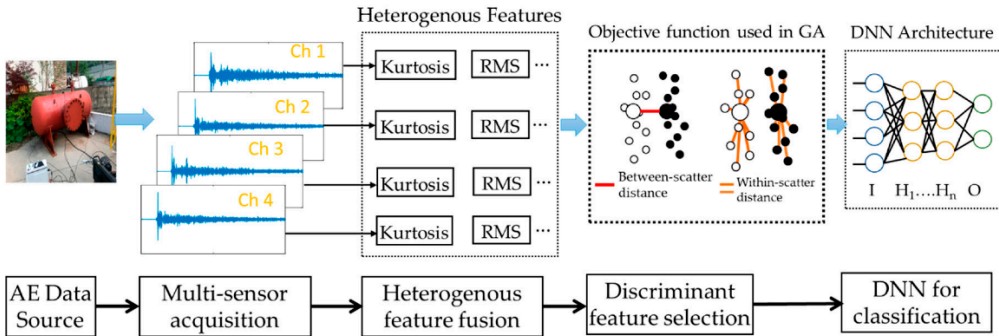

**Figure 29.** Diagram of the proposed method for classifying pressure vessel cracks [111].

Several intelligent optimization algorithms that imitate the evolution principles of biological computing disciplines, artificial intelligence, and immunology are currently being successfully used in composite material design. Kim et al. [112] used a semi-geodesic path algorithm, a progressive failure analysis, and a modified genetic algorithm to optimize a Type 3 (aluminum-walled, fiber-reinforced) vessel subjected to internal pressure. The primary goal was to reduce vessel weight while preventing failure. This work extended using a semi-geodesic path method, finite element (FE), and a genetic algorithm to reduce vessel weight [113].

Giordano et al. [114] devised an optimization strategy based on the Finite Element Method (FEM) to reduce bulk while boosting stiffness. Liu et al. [115] employed the artificial immune system (AIS) technique to reduce the weight of a Type 3 hydrogen storage vessel. The weight of a composite hydrogen storage vessel subjected to burst pressure was reduced by Xu et al. [116]. To optimize the vessel, they presented an adaptive genetic algorithm. Their method's performance was compared to a simple genetic algorithm and a Monte Carlo optimization method.

Machine learning has been used to predict the mechanical properties of different materials [117–121] for composite materials. A computational model based on machine learning, sometimes called a surrogate model, can have varied methods dependent on algorithm architecture, dataset size, and required processing power. It boils down to the model's final functionality and associated proven qualities. ML algorithms are categorized as supervised or unsupervised learning depending on their method of operation. Figure 30 depicts a categorization system highlighting several types and instances of common ML methods. The supervised learning strategy utilizes a function that maps labeled training datasets as input, which may be applied to predict the label of unlabeled examples.

The ML algorithm derives a function from training data and maps it onto new output data. Regression and classification issues are characterized as supervised learning challenges. Regression is concerned with quantitative labeling, such as estimating the number of instances of an object in a given image. Classification is concerned with qualitative data, such as indicating whether an image is of a specific type. Linear regression, logistic regression, neural networks, multilayer perceptron, support vector machines, naive Bayes, decision trees, and the K-nearest neighbor algorithm are the most prominent learning algorithms.

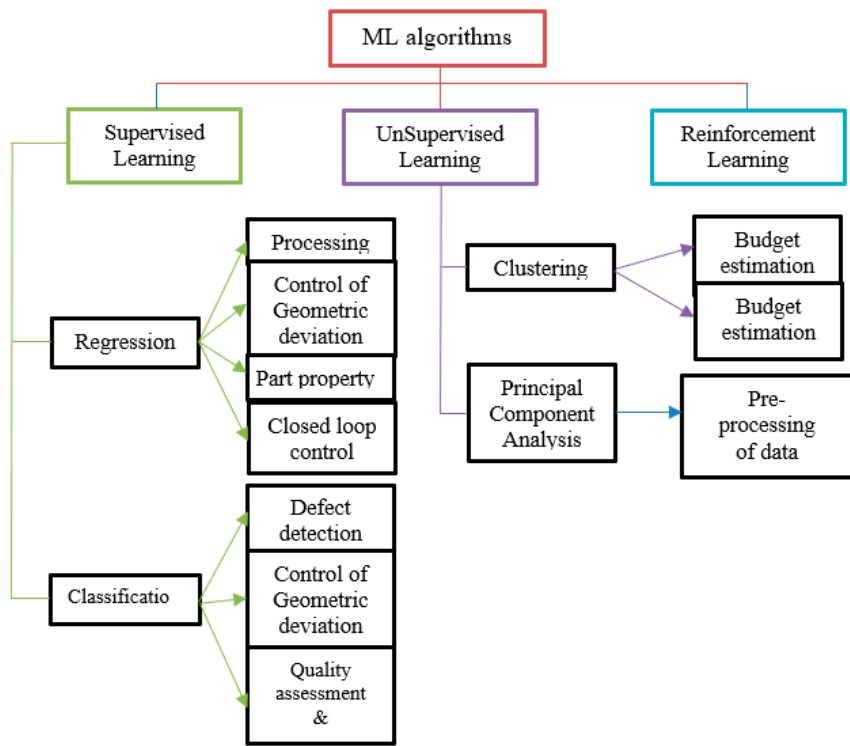

**Figure 30.** A classification scheme of machine learning algorithms based on their operating principle.

Figure 31 depicts the basic structure of an ML process, which includes the preparation (also called cleaning) of the input data from multiple sources, the selection of suitable descriptors for appropriate representation, the selection of the algorithm, and, finally, the use of the developed model for prediction and further applications. Erroneous information must be removed; therefore, data cleaning is a must, and after this comes normalization, standardization, and stratified sampling to deal with discrepancies. Related hyper-parameters must be optimized whenever there is a change in the data. In order to derive conclusions based on domain knowledge and test models, it is necessary to use unseen data for testing and cross-validation. Input data for a machine learning model can come from various places, including preexisting databases, published articles, high-throughput simulations, and experiments. The data connects the utilized ML model and the investigation's underlying problem mechanics. To ensure the maximum effectiveness of the ML model, the primary difficulties here are around collecting enough high-quality data.

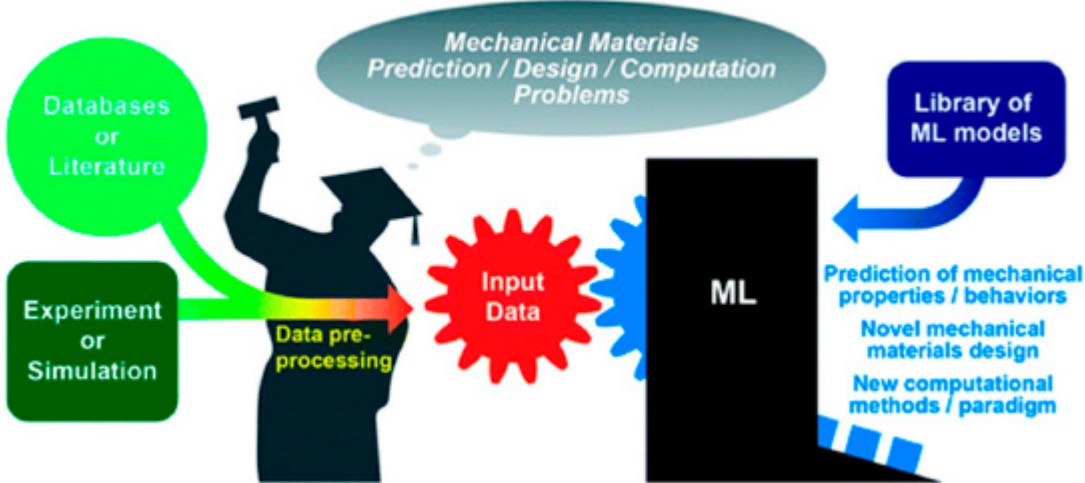

**Figure 31.** A typical workflow of an ML application in a material design problem [35].

Hu et al. [122,123] used data-driven methods for various composite materials and made a robust alternative computing tool for analyzing composite structures. Gu et al. [124] used machine learning on a composite system and showed that it could accurately and effectively predict mechanical properties, such as toughness and strength. This gives designers a new way to think about design and speeds up the process of finding new functional composites that can be customized. Kaveh et al. [125] developed an optimal design strategy based on machine learning to predict the ultimate buckling load of cylindrical composite materials with different stiffness. The neural network was trained with 11,000 samples and got good results. Using artificial neural networks, Luo Ling et al. [117] significantly improve the accuracy and efficiency of predicting how composite materials will distort. This helps guide the design of composite structures, such as asymmetrical laminates, in a good way. As far as we know, machine learning has proven to be a valuable tool for designing materials and can accurately predict how they will behave. Compared to traditional methods, machine learning approaches will take much less time to do computations.

Artificial Neural Networks (ANN) can be employed in manufacturing hydrogen storage vessels [126] due to their compatibility with big datasets and the practical inference of non-linear decision boundaries. However, as depicted in Figure 32, the mapping between input and output spaces may consist of a sequence of processing units known as hidden layers of varying dimensions. CNNs designed for specific applications, such as computer vision, are also created. In computer vision tasks, CNNs explore the input space to detect comparable features, such as vertical lines. Also defined as deep learning are neural networks with more than two hidden layers (DNN). Hidden layers multiply the input layer's weight by a factor and add a bias based on the output layer.

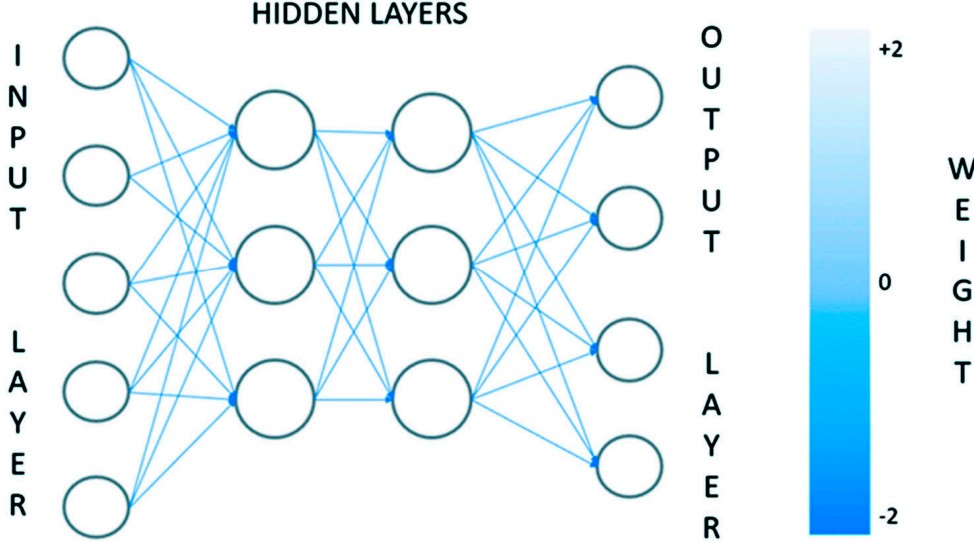

**Figure 32.** Schematic of an ANN architecture [127].

Azizian and Almeida Jr (2022) [128] used numerical simulations and artificial neural network (ANN) metamodels to investigate the probabilistic behavior and reliability of internally-pressurized filament wound composite tubes (Figure 33). The authors developed ANNs to predict the behavior of these tubes under various loading conditions and then used these metamodels to perform stochastic and probabilistic analyses. Through these numerical investigations, the authors gained insights into the behavior of composite tubes and their reliability under uncertain loading conditions.

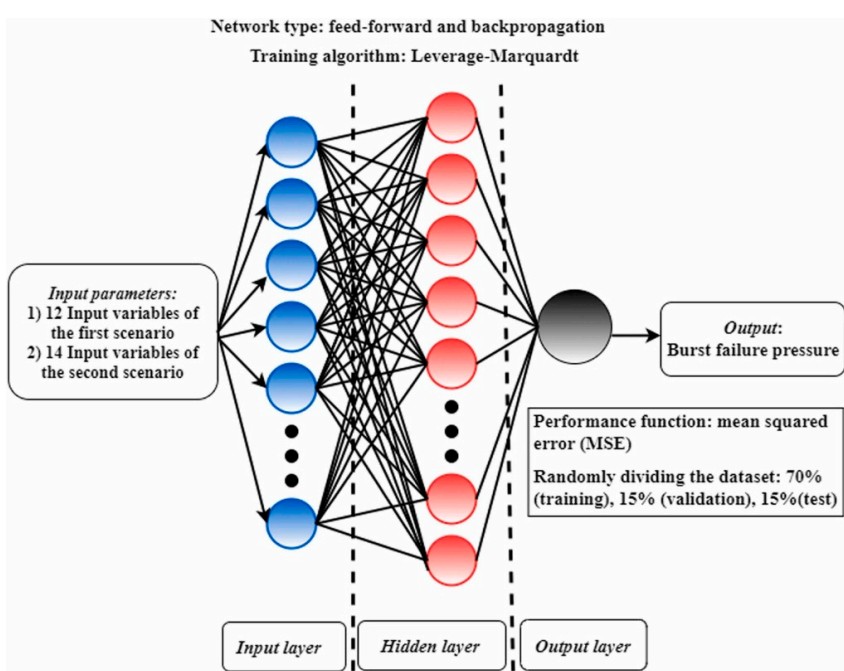

**Figure 33.** The architecture and details of the constructed ANN [128].

There have been several research works that have used artificial intelligence (AI) models for the design of hydrogen storage vessels. One of the main challenges in designing these vessels is the need to optimize the storage capacity while minimizing the weight and volume of the vessel. This requires a detailed analysis of the physical and chemical properties of the materials used for the vessel, as well as design constraints such as temperature and pressure [129]. In paper [130], a genetic algorithm (GA) to optimize the design of a hydrogen storage vessel was made of a metal hydride material. The GA identified the optimal combination of material and design parameters that resulted in the highest storage capacity. The flow chart of the adaptive genetic algorithm (AGA) is shown in Figure 34. The number of generations (EN) was used to control the solution cycle. In each generation, the fitness value, the weight and burst pressure of the composite vessel, and the probabilities $P_c$ and $P_m$ of crossover and mutation for each individual were calculated [116]. Another approach is using machine learning algorithms, such as neural networks and support vector machines (SVMs), to predict the storage capacity of hydrogen storage vessels. For example, Kim et al. [131] used a neural network to predict the storage capacity of a vessel made of a carbon nanotube material, while in paper [132], presents a method for predicting hydrogen storage capacity of V-Ti-Cr-Fe alloy using ensemble machine learning. In addition to machine learning algorithms, researchers [128] have also used optimization algorithms, such as the particle swarm optimization (PSO) algorithm, to optimize the design of hydrogen storage vessels.

Recently, researchers [133] found that the AI-based approach improved performance and cost-effectiveness compared to traditional design methods, which used a combination of artificial neural networks and evolutionary algorithms to optimize the design of high-pressure hydrogen storage vessels.

The development of a digital twin for a composite hydrogen storage vessel involves creating a computational model that can predict the behavior and performance of the physical vessel, without requiring the need for expensive and time-consuming full-scale prototyping and testing. The digital twin uses available information and data to accurately forecast the future performance of the hydrogen storage vessel. This approach can help optimize the design and performance of the hydrogen storage vessel, ensuring its safety and reliability. Burov and Burova (2020) [134] focused on the development of a digital twin for a composite overwrapped pressure vessel (COPV) commonly used in electric

propulsion engines for spacecraft (Figure 35). The purpose of the digital twin was to accurately predict the future behavior and performance of the physical COPV, without the need for expensive and time-consuming full-scale prototyping and testing. The authors described the process of developing the digital twin, including the selection of appropriate modeling and simulation techniques. The article highlighted the potential of digital twins for optimizing the design and performance of composite pressure vessels used in space applications, and the validation of the developed digital twin through experimental data was also discussed.

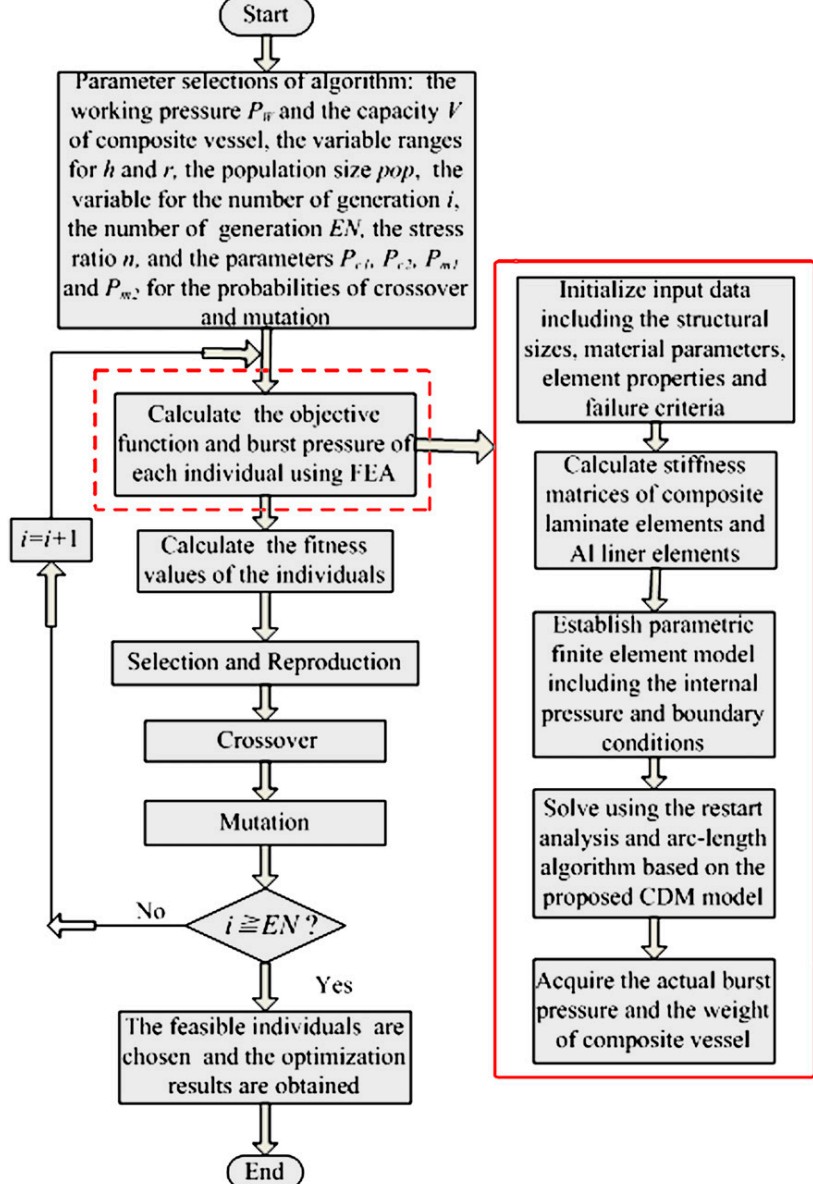

**Figure 34.** Flow chart of the adaptive genetic algorithm [116].

Hopmann et al. (2022) [135] investigated the effect of winding parameters on the fiber bandwidth in the cylindrical area of a hydrogen pressure vessel. The purpose of the study was to generate a digital twin of a composite hydrogen storage vessel that could predict its future behavior and performance. The article focused on the impact of winding parameters, such as winding angle, tension, and speed, on the fiber bandwidth in the cylindrical area of the pressure vessel. The authors used computer simulations and experimental data to determine the optimum winding parameters for achieving a uniform fiber distribution, which was crucial for the performance and safety of the hydrogen storage vessel. The study highlighted

the importance of accurate modeling and simulation of the manufacturing process to develop a reliable digital twin for composite hydrogen storage vessels (Figure 36). Overall, the article provided valuable insights into the development of a digital twin for composite hydrogen storage vessels and could aid in the optimization of their design and performance.

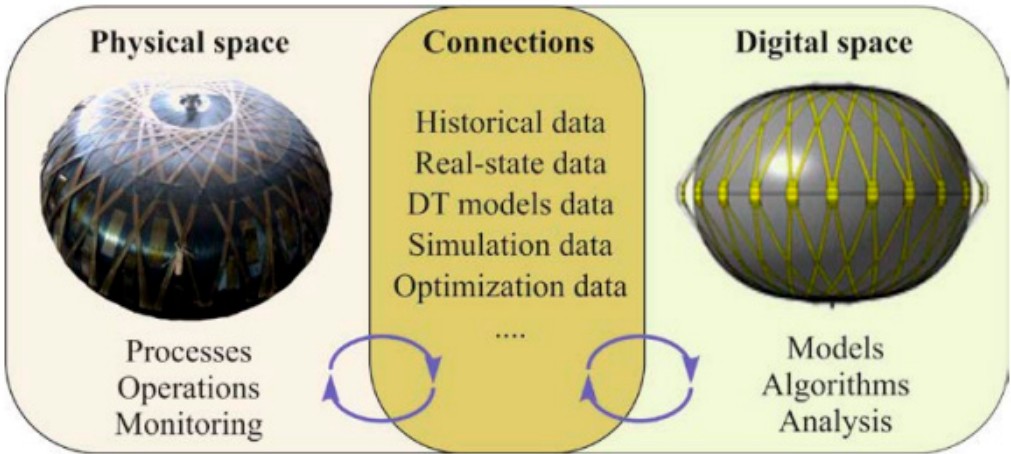

**Figure 35.** Concept of digital twin for composite pressure vessels [134].

Jaribion, A. et al. [136] developed a prototype consisting of hardware and software components to create a digital twin for industrial application. The prototype was aimed at enabling the creation of a digital twin for high-pressure hydrogen vessels, which are industrial equipment with a high safety requirement for the storage and transfer of highly flammable hydrogen. The effectiveness of utilizing a real-time digital twin of the hydrogen high-pressure vessel for failure risk management was demonstrated by the prototype. Action Design Research (ADR) was used to describe the process that led to the development of the prototype (Figure 37).

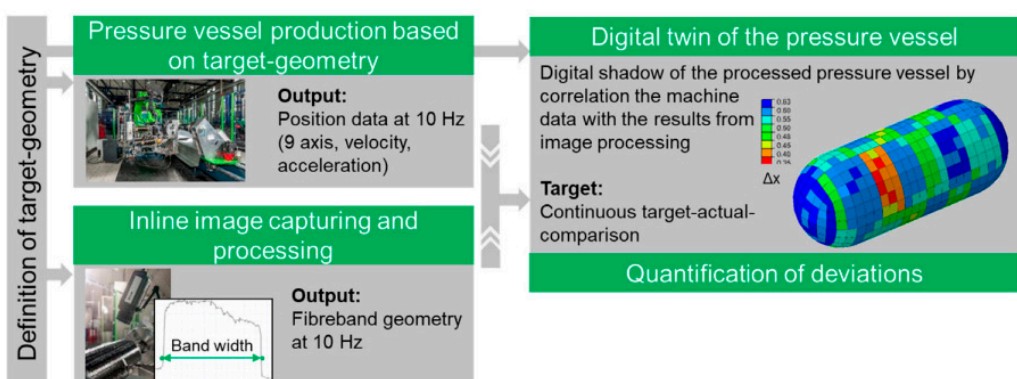

**Figure 36.** Digital twin during manufacturing of Type IV pressure vessels for design optimization [135].

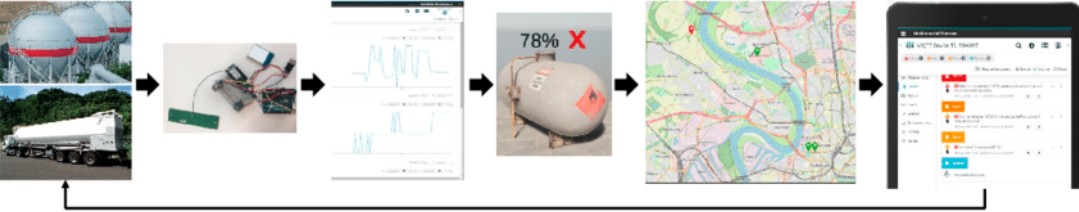

**Figure 37.** A digital twin for safety and risk management: a hydrogen high-pressure vessel case [136].

## 7. Discusion

Composite materials, such as carbon fiber reinforced polymer (CFRP), are increasingly being used in the construction of hydrogen storage vessels due to their lightweight, high-strength, and corrosion-resistant properties. However, designing and testing these vessels can be expensive and time-consuming, making it difficult to quickly iterate and improve upon designs. Artificial intelligence (AI) is another important area of research in hydrogen storage vessel design. AI algorithms can be used to analyze large amounts of data collected from sensors on a hydrogen storage vessel, allowing engineers to identify patterns and optimize the design and operation of the vessel. This is where digital twin technology comes in. A digital twin is a virtual model of a physical object or system that uses real-time data and simulations to predict its behavior and performance. By creating a digital twin of a hydrogen storage vessel, engineers can test and optimize designs in a virtual environment before building a physical prototype. This can significantly reduce the cost and time required for design and testing.

While hydrogen storage vessels made of composite materials offer many advantages over traditional metal vessels, there are still some challenges that need to be addressed. For example, composite materials have different failure modes than metals, and predicting their behavior under different loading conditions can be challenging. Figure 38 showed the presence of numerous matrix cracks (Figure 38a,b) at the cylinder dome conjunction with the composite pressure vessel under a pressure of 44 MPa [92].

There are also some potential disadvantages of using artificial intelligence (AI) and digital twin technology in the development of hydrogen storage vessels. One potential disadvantage is that the use of AI requires large amounts of data, and the quality and accuracy of the data can greatly affect the reliability and effectiveness of the AI model. This can be a challenge in the case of hydrogen storage vessels, where data on their behavior and performance may be limited and difficult to obtain. Another disadvantage is the potential for the AI model to be biased, either due to the quality of the data used or due to the inherent biases of the AI algorithms themselves. This can lead to inaccurate predictions or recommendations, and could have serious consequences in the case of safety-critical systems such as hydrogen storage vessels. Additionally, digital twin technology requires significant computational resources to simulate the behavior and performance of the physical system in real time. This can be a challenge for industrial applications, where real-time performance is critical and there may be limited resources available for computing. Finally, there is the risk of cyber attacks on the digital twin system, which could compromise the safety and security of the physical system it is modeling. It is important to implement strong cybersecurity measures to protect against these risks.

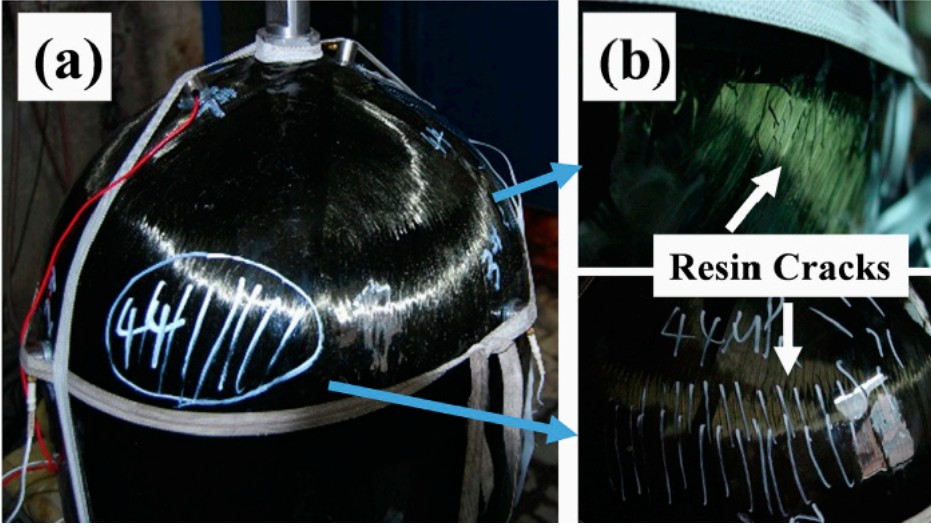

**Figure 38.** Numerous matrix cracks at the dome of the composite vessel [92].

## 8. Conclusions

Composite hydrogen storage vessels have been the subject of numerous research studies due to their potential use in various applications, such as hydrogen fuel cells for transportation and stationary power generation. These vessels, which typically consist of a combination of materials such as metal hydrides and carbon nanotubes, offer several advantages over traditional storage methods, including higher capacity, lower weight, and improved safety. To better understand and optimize the performance of these vessels, several researchers have employed a variety of modeling approaches at multiple scales. These include molecular dynamics simulations to study the interactions between hydrogen and the storage materials at the atomic level, as well as continuum-based models that consider the overall mechanical behavior of the vessel under various loading conditions.

One key aspect of these modeling efforts has been the development of constitutive models that accurately capture the behavior of the composite materials under different loading and temperature conditions. These models have been used to predict the amount of hydrogen that can be stored in the vessel at different pressures and temperatures and the rate at which the hydrogen is absorbed and released. Other research works have focused on understanding the microstructural features of the composite materials that are most effective at enhancing hydrogen storage capacity. In addition to these microscale studies, researchers have also employed macroscale models to predict the overall behavior of the composite hydrogen storage vessels under different loading conditions. These models have been used to optimize the design and performance of the vessels, including the shape, size, and material selection. Overall, multi-scale modeling has been critical in advancing our understanding of composite hydrogen storage vessels and developing improved designs for various applications. Composite materials have been widely used in the design and construction of hydrogen storage vessels due to their high strength and low weight. These materials combine two or more materials, such as fiber-reinforced plastics or metals, which can be tailored to meet specific performance requirements. Recent advances in composite materials have focused on improving their mechanical properties, thermal stability, and corrosion resistance.

Artificial intelligence (AI) has also been used to design hydrogen storage vessels in order to improve their performance and make them more efficient. AI algorithms can be used to predict the behavior of materials under different conditions and optimize the vessel's design based on these predictions; this allows for optimizing the vessel's size, shape, and thickness to meet the application's specific requirements. One example of using composite materials and AI in hydrogen storage vessels is the development of high-pressure composite vessels for use in fuel cell vehicles. These vessels are designed to store high pressures of hydrogen gas safely and efficiently, and their design is optimized using AI algorithms to minimize weight and maximize performance. Overall, the combination of composite materials and AI in hydrogen storage vessel design has the potential to improve these systems' efficiency and performance significantly and has already led to the development of more advanced and cost-effective solutions for hydrogen storage.

**Author Contributions:** M.N.: Methodology, Investigation, Writing—original draft, Writing—review & editing, Conceptualization, Data curation, Supervision, Validation, Visualization. M.T. and M.R.: Investigation, Methodology, Writing—original draft, Writing—review & editing; M.a.A., A.V., A.A., H.L., H.M., H.N.: Validation, Visualization; All authors have read and agreed to the published version of the manuscript.

**Funding:** This research received no external funding.

**Data Availability Statement:** The data presented in this study are available on request from the corresponding author.

**Acknowledgments:** We would like to express our sincere gratitude to Lee Harper from university of Nottingham for their invaluable contributions to this project. Their rigorous review and insightful comments have significantly strengthened the quality of our research. Their engagement with our work has helped us to refine our arguments and clarify our conclusions. We are deeply appreciative of their time, effort, and expertise, and we believe that their contributions have made a significant impact on our manuscript.

**Conflicts of Interest:** The authors declare no conflict of interest. The funders had no role in the design of the study; in the collection, analysis, or interpretation of data; in the writing of the manuscript, or in the decision to publish the results.

## Nomenclature

| | |
|---|---|
| M4H2 | Multi-scale modeling, Multi-physics, Multi-scale experiments; and Machine learning for hydrogen storage vessel design |
| AGA: | Adaptive Genetic Algorithm |
| AI: | Artificial Intelligence |
| AIS: | Artificial Immune System |
| AM: | Additive Manufacturing |
| AMMC: | Aluminum Metal Matrix Composite |
| ANN: | Artificial Neural Network |
| $CcH_2$: | Cryo-Compressed Hydrogen Storage |
| CDM: | Continuum Damage Mechanics |
| CE: | Cohesive Element |
| CFD: | Computational Fluid Dynamics |
| CFRP: | Carbon Fiber Reinforced Plastic |
| $CGH_2$: | Compressed Hydrogen Storage |
| CLT: | Classical Laminate Theory |
| CNN: | Convolutional Neural Network |
| CNT: | Carbon Nanotube |
| COPV: | Composite Overwrapped Pressure Vessel |
| DNN: | Deep Neural Network |
| DOE: | Department Of Energy |
| FEA: | Finite Element Analysis |
| FEM: | Finite Element Method |
| GA: | Genetic Algorithm |
| GFRP: | Glass Fiber Reinforced Plastic |
| HSV: | Hydrogen Storage Vessel |
| $LH_2$: | Cryogenic Hydrogen Storage |
| ML: | Machine Learning |
| MMC: | Magnesium Metal Matrix Composite |
| PEG: | Polyethylene Glycol |
| PEM: | Proton Exchange Membrane |
| PSO: | Particle Swarm Optimization |
| RVE: | Representative Volume Elements |
| SA: | Simulated Annealing |
| SSC: | Stainless Steel Composite |
| SVM: | Support Vector Machines |
| TMMC: | Titanium Metal Matrix Composite |

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
