# Peer review of "An Overview of the Recent Advances in Composite Materials and Artificial Intelligence for Hydrogen Storage Vessels Design"

_jcs, doi:10.3390/jcs7030119_

Round 1

Reviewer 1 Report

The attractive review paper focuses on composite material and AI for hydrogen storage vessel design. It can be published with the address of the following issues. 

(1) There are many abbreviations in the whole paper. I suggest summarizing all the abbreviations in a nomenclature table. 

(2) Some H2 put 2 at the subscript and some not. It should be consistent. 

(3) The review of CFRP is too general. The authors should review the current CFRP  first, like the effects of T400 to T1000 on the cost and performance of hydrogen storage vessels. With the analysis of current material, the readers can understand why the application of new material (graphene and CNT) is important. Though the authors mention several advantages of CNT, the given information is too general. The readers would like to see the percentage of saved weight and increasing strength of storage. Moreover, the effects of CNT's length and types may also need to be discussed.

(4) The review of AI to the hydrogen storage vessel is too general. The authors should connect AI with design more tightly. The following questions can serve as a reference for the authors to revise this part:

(4.1)  The comparison of supervised learning and unsupervised learning. The available data for the hydrogen storage design is not that much. Therefore unsupervised learning may not be necessary. 

(4.2) Where is the source of the training set for the hydrogen storage design? And how to select the testing set?

(4.3) How to determine and select the input features for the supervised ML?

(4.4) The authors introduce ANN application in the hydrogen storage vessel design. I don't understand why the authors suddenly mentioned computer vision (CV). Is CV used in this work?  The authors also discussed hidden layer. Why do we need the hidden layers in the ANN for hydrogen storage vessel design?

Author Response

Respected  Reviewer!

We greatly appreciate the valuable comments made by the reviewer. These comments are precious and helpful for revising and improving our paper and guiding future research.

After carefully studying the comments, we have made corresponding changes highlighted in red font color in the revised manuscript. The followings are the answers and revisions we have made in responding to the reviewers' comments on an item-by-item basis.

  • Reviewer 1

The attractive review paper focuses on composite material and AI for hydrogen storage vessel design. It can be published with the address of the following issues.

  • Question 1

There are many abbreviations in the whole paper. I suggest summarizing all the abbreviations in a nomenclature table.

  • Answer 1

Thank you for your feedback. We appreciate your suggestion to include a nomenclature table summarizing all the abbreviations used in the paper. We agree that this makes the paper easier to read and understand, especially for readers who may not be familiar with all the abbreviations used.

We have incorporated your suggestion and included a nomenclature table in the paper. This table lists all the abbreviations used in the paper, their complete forms, and brief explanations of their meanings.

  • Question 2

Some H2 put 2 at the subscript and some not. It should be consistent. 

  • Answer 2

Thank you for bringing this to our attention. We apologize for any confusion caused by the inconsistent use of subscripts for the "H2" notation in the paper. We agree that it is essential to maintain consistency throughout the manuscript to avoid any ambiguity.

We carefully review the manuscript and make sure that all instances of "H2" notation are standardized.

  • Question 3

The review of CFRP is too general. The authors should review the current CFRP  first, like the effects of T400 to T1000 on the cost and performance of hydrogen storage vessels. With the analysis of current material, the readers can understand why the application of new material (graphene and CNT) is important. Though the authors mention several advantages of CNT, the given information is too general. The readers would like to see the percentage of saved weight and increasing strength of storage. Moreover, the effects of CNT's length and types may also need to be discussed.

  • Answer 3

Thank you for your valuable feedback. We appreciate your suggestions on improving the manuscript, especially in providing more detailed information on the current state of Carbon Fiber Reinforced Polymer (CFRP) and the potential benefits of using graphene and carbon nanotubes (CNTs) as reinforcements.

We carefully review the manuscript and make the necessary revisions to address your concerns. Specifically, we review the current literature on the effects of T400 to T1000 on the cost and performance of hydrogen storage vessels and include a more in-depth analysis of the current material. We also provide more specific information on the advantages of using CNTs, including the percentage of saved weight and increased storage strength, and discuss the effects of CNT length and types.

  • Question 4.1

The review of AI to the hydrogen storage vessel is too general. The authors should connect AI with design more tightly. The following questions can serve as a reference for the authors to revise this part:

The comparison of supervised learning and unsupervised learning. The available data for the hydrogen storage design is not that much. Therefore unsupervised learning may not be necessary. 

  • Answer 4.1

I agree with the reviewer's feedback that the review of artificial intelligence (AI) in the context of hydrogen storage vessel design is too general. While AI has the potential to revolutionize the field of hydrogen storage technology, there is still limited research and practical implementation of AI methods for designing hydrogen storage vessels. Therefore, it is important to conduct more specific and focused studies to explore the potential applications of AI in this field.

The use of AI in hydrogen storage vessel design is a rapidly evolving field, and there is a growing body of research exploring different approaches to using AI to improve the performance and efficiency of hydrogen storage systems. In response to your comment on connecting AI with design more tightly, we agree that a more specific discussion of AI in the context of hydrogen storage vessel design would be beneficial. We revise our manuscript to include more concrete examples and practical applications of AI in the design process.

Regarding your suggestion for comparing supervised and unsupervised learning, we agree this could be a valuable addition to our manuscript. We explore this topic in more detail and discuss each approach's potential advantages and limitations, particularly in the context of the limited available data for hydrogen storage vessel design.

  • Question 4.2

Where is the source of the training set for the hydrogen storage design? And how to select the testing set?

  • Answer 4.2

Thank you for your question regarding the source of the hydrogen storage design training set and the testing set selection. We appreciate your interest in our research and are happy to provide more information on these topics.

The source of the training set for hydrogen storage vessel design will depend on the specific problem statement and available data. In general, the training set can be sourced from experimental or simulation data, or from existing datasets in the literature or industry.

If experimental data is available, it can be used to train a machine learning model to predict the behavior of hydrogen storage vessels under different conditions. This can involve testing the vessels under a range of pressures, temperatures, and other relevant variables, and collecting data on their performance. Alternatively, simulation data can be generated using software tools that simulate the behavior of hydrogen storage vessels based on input variables such as material properties, geometrical features, and operating conditions.

Once the training set is selected, the testing set can be chosen using a range of techniques. One common approach is to randomly partition the dataset into a training set and a testing set, with a majority of the data used for training and a smaller subset used for testing the accuracy and generalization of the model. Another approach is to use cross-validation, where the dataset is divided into several folds and the model is trained and tested on each fold in turn.

It's important to ensure that the testing set is representative of the data that the model is expected to encounter in real-world scenarios. This means that the testing set should include a diverse range of data points, and should not contain any data points that were used in the training set. By carefully selecting and partitioning the training and testing sets, you can ensure that your machine learning model is accurate and reliable for predicting the behavior of hydrogen storage vessels.

  • Question 4.3

How to determine and select the input features for the supervised ML?

  • Answer 4.3

The selection of input features is a crucial step in supervised ML, as it directly affects the performance and generalization capability of the AI models. In general, the input features should be relevant, informative, and independent of each other, as correlated or redundant features can introduce noise and bias into the model. To develop a supervised machine learning model for hydrogen storage vessel design, the input features should be relevant, informative, and independent of each other, as correlated or redundant features can introduce noise and bias into the model. Some potential input features that could be considered include material properties, geometrical features, pressure and temperature requirements, safety factors, hydrogen storage capacity, operating conditions, cost, maintenance and repair requirements, environmental impact, and other relevant factors specific to the problem statement.

  • Question 4.4

The authors introduce ANN application in the hydrogen storage vessel design. I don't understand why the authors suddenly mentioned computer vision (CV). Is CV used in this work?  The authors also discussed hidden layer. Why do we need the hidden layers in the ANN for hydrogen storage vessel design?

  • Answer 4.4

Thank you for your question regarding our introduction of computer vision (CV) and using hidden layers in artificial neural networks (ANNs) for hydrogen storage vessel design. We appreciate your interest in our research and would like to clarify these points.

Regarding the mention of CV, we apologize for any confusion that may have arisen. This paper focuses on applying ANNs for hydrogen storage vessel design, and CV is not directly used in our work. The mention of CV was intended to provide a broader context for the potential applications of AI in materials science and engineering, including CV for image-based analysis of materials and structures.

Regarding the use of hidden layers in ANNs for hydrogen storage vessel design, these layers are necessary to allow the model to learn and represent complex nonlinear relationships between the input and output variables. Hidden layers consist of nodes or neurons that use nonlinear activation functions to transform the input signals and generate the output signals. The number and size of the hidden layers depend on the complexity and size of the input and output spaces and the desired performance of the model.

In hydrogen storage vessel design, the input space may include parameters such as material properties, geometric parameters, operating conditions, and safety factors, while the output space may include performance metrics such as hydrogen storage capacity, release rate, stability, and cost. The hidden layers allow the ANN to capture the nonlinear interactions and dependencies between these input and output variables, which may not be easily modeled using linear or simple mathematical functions.

Reviewer 2 Report

Referee report about

"An Overview of the Recent Advances in Composite Materials and Artificial Intelligence for Hydrogen Storage Vessels Design"

by M. Nachtane et al.

The manuscript is a review about technological developments in the use of composite materials and computational techniques for the design of hydrogen storage vessels (HSVs).
The review gives an overview about hydrogen storage technologies.
The authors discuss design theories, composite materials used in HSVs and the use of finite element methods and artificial intelligence in the design process.

Some of the sections are too general and only touch the surface. I have given a longer list of comments below. Most of the figures seem to be copied from other articles. The situation concerning the copyrights should be solved.

To summarize, the paper is suitable for publication in the Journal of Composites Sciences.

However, the authors should modify the paper following the comments below.

Comments on Section 2:

It would be nice, if the authors could give a quantitative comparison of the different storage systems. For example: Could the costs per liter hydrogen and weight per liter hydrogen compared in a plot or a table?

Comments on Section 5:

The authors should explain in more detail the finite element analysis (FEA), Multi-scale modelling and computational fluid dynamics.
1.) What are the assumptions and limitations?
2.) What are the computational costs? Which software packages are available?
3.) What can be calculated with respect to composite materials and hydrogen storage vessels?
4.) How big are the errors of the model and numerics?
5.) What is the difference between meso-macro and micro-meso-macro approaches?

Comments on Section 6:

The authors should more clearly point out the following points:
1.) What are the objective functions, that are used in the optimization?
2.) Which parameters of the vessels are optimized?
3.) Which successes have been done in the design of the vessels using AI?

Additional comments:

Most of the Figures seem to be copied from the literature.
A citation should be given in the caption of the figure.
As already mentioned above, the authors should check the situation concerning the copyrights of all Figures.

For example:
Figure 2 on page 4: reference [37] ?
Figure 3 on page 5: reference [38] ?
Figure 7 on page 9: reference ???
Figure 10 on page 12: reference [67] ?
Figure 11 on page 13: reference ???
Figure 12 on page 14: reference ???
Figure 14 on page 16: reference ???
Figure 17 on page 19: reference ???
Figure 18 on page 20: reference ???
Figure 19 on page 21: reference ???
Figure 21 on page 22: reference ???

last sentence on page 10:
"Some common materials used in the design of composite hydrogen storage vessels include:"
The text continues with Figure 8

page 12, line 400:
"Recently, the researchers investigate ... "
Which researchers? - please add a citation.

pages 15 - 17:
The abbreviation for finite element analysis (FEA) is introduced 5 times on the pages 15 - 17.
This can be reduced to one time.

On page 16, line 551:
"Another researcher, the authors reviewed existing ..."
Please add some name and reference.

On page 18:
references for ABAQUS and ANSYS

On page 19 (line 606):
What is the failure criterion?

On page 26 (lines 832 - 834):
"In addition to machine learning algorithms, researchers ... "
Which researchers? please give a reference.

Author Response

Respected Reviewer!

We greatly appreciate the valuable comments made by the reviewer. These comments are precious and helpful for revising and improving our paper and guiding future research.

After carefully studying the comments, we have made corresponding changes highlighted in red font color in the revised manuscript. The followings are the answers and revisions we have made in responding to the reviewers' comments on an item-by-item basis.

Reviewer 2

Referee report about

"An Overview of the Recent Advances in Composite Materials and Artificial Intelligence for Hydrogen Storage Vessels Design"

by M. Nachtane et al.

The manuscript is a review about technological developments in the use of composite materials and computational techniques for the design of hydrogen storage vessels (HSVs).

The review gives an overview about hydrogen storage technologies.

The authors discuss design theories, composite materials used in HSVs and the use of finite element methods and artificial intelligence in the design process.

Some of the sections are too general and only touch the surface. I have given a longer list of comments below. Most of the figures seem to be copied from other articles. The situation concerning the copyrights should be solved.

To summarize, the paper is suitable for publication in the Journal of Composites Sciences.

However, the authors should modify the paper following the comments below.

  • Question 1

Comments on Section 2:

It would be nice, if the authors could give a quantitative comparison of the different storage systems. For example: Could the costs per liter hydrogen and weight per liter hydrogen compared in a plot or a table?Thank you for your insightful feedback. We appreciate your suggestion to include a quantitative comparison of the various storage systems, as well as the weight and cost per liter of hydrogen.

We have added additional information to the revised version.

  • Answer 1

Thank you for your insightful feedback. We appreciate your suggestion to include a quantitative comparison of the various storage systems, as well as the weight and cost per liter of hydrogen. We have added additional information to the revised version.

  • Question 2

Comments on Section 5:

The authors should explain in more detail the finite element analysis (FEA), Multi-scale modelling and computational fluid dynamics.

Finite Element Analysis (FEA): FEA is an engineering simulation technique used to predict the behavior of structures under loads and environmental conditions. FEA allows engineers to analyze a hydrogen storage tank's structural integrity and performance under various conditions. In FEA, the structure is divided into several small finite elements, and the motion and boundary conditions equations are applied to the elements. FEA can then be used to obtain stress and strain distributions, natural frequencies, and other structural properties.

Multi-Scale Modelling: Multi-scale modelling is an engineering simulation technique used to analyze how different materials interact with each other over different size scales. This technique can simulate the material properties of hydrogen storage tanks at the macroscopic and microscopic levels. For example, multi-scale modelling can be used to simulate the material properties of the tank walls and analyze how the wall material responds to changes in pressure, temperature, and other conditions.

Computational Fluid Dynamics: Computational fluid dynamics (CFD) is an engineering simulation technique used to analyze how fluids and gases interact with each other. CFD can be used to simulate the behavior of hydrogen gas in a storage tank and the effects of different parameters on the tank's internal pressure. CFD can also be used to analyze the effects of pressure and temperature on the gas flow and the failure mechanisms of the tank.

1.) What are the assumptions and limitations ?

Finite Element Analysis (FEA)

Assumptions:

- FEA assumes that the material behaves as a continuum, i.e. it is infinitely divisible and that the behavior of any part of the material can be expressed by the behavior of the whole material.

Limitations:

- FEA relies heavily on the accuracy of the input data and the assumptions made about material behavior.

- FEA is limited by the number of elements used in the simulation, as the accuracy of the results depends on the resolution of the model.

Assumptions:

  1. The material properties of the tank and its components are homogeneous and isotropic.
  2. The stress-strain behavior of the tank material is linear elastic.
  3. The deformation of the tank is small compared to its size.
  4. Forces and moments applied to the tank are distributed uniformly over its surface.
  5. The pressure and temperature of the hydrogen stored in the tank remain constant.

Limitations:

  1. FEA cannot predict the fatigue life of the tank.
  2. FEA cannot accurately predict the behavior of non-homogeneous or anisotropic materials.
  3. FEA cannot accurately predict the behavior of a structure that undergoes large deformations.
  4. FEA cannot accurately predict the behavior of structures under dynamic loading.
  5. FEA is not suitable for predicting the behavior of structures subject to non-linear material behavior.

Multi-scale Modelling

Assumptions:

- Multi-scale modelling assumes that the material can be broken down into its constituent parts and that the behavior of the material can be described by the interactions between these parts.

Limitations:

- Multi-scale modelling is limited by the resolution of the model, as the accuracy of the results depends on the accuracy of the input data.

- Multi-scale modelling is also limited by the complexity of the problem, as more complex problems require more complex models.

Assumptions:

  1. The model assumes that the hydrogen is stored in a large tank and the flow of hydrogen is predominantly laminar.
  2. The model assumes that the temperature and pressure of the hydrogen are constant throughout the tank.
  3. The model assumes that the flow of hydrogen is steady and uniform across the entire tank.

Limitations:

  1. The model does not account for turbulence or other non-uniform flow characteristics which may lead to pressure drops and localized temperature changes.
  2. The model does not account for the effects of heat transfer and the associated temperature gradients which can occur in large hydrogen tanks.
  3. The model does not take into account the presence of hydrogen impurities which may affect the performance of the tank.

Computational Fluid Dynamics

Assumptions:

- CFD assumes that the material behaves as a continuum, i.e. it is infinitely divisible and that the behavior of any part of the material can be expressed by the behavior of the whole material.

Limitations:

- CFD is limited by the accuracy of the input data and the assumptions made about material behavior.

- CFD is also limited by the complexity of the problem, as more complex problems require more complex models.

Assumptions:

  1. Ideal gas law is used to model the behavior of the hydrogen gas inside the storage tank.
  2. The pressure, temperature, and density of the hydrogen gas are uniform throughout the tank.
  3. The flow of hydrogen gas is assumed to be laminar and incompressible.

Limitations:

  1. Complex geometries of the storage tank can create turbulence in the flow of hydrogen gas, which is not taken into account in Computational Fluid Dynamics models.
  2. Heat transfer is not taken into account, which can affect the pressure and temperature of the hydrogen gas inside the storage tank.
  3. The effects of chemical reactions on the hydrogen gas are also not considered in Computational Fluid Dynamics models.

2.) What are the computational costs ? Which software packages are available ?

The computational costs of finite element analysis (FEA), multi-scale modeling, and computational fluid dynamics (CFD) can be significant, especially when dealing with complex systems like hydrogen storage vessels. These methods involve solving partial differential equations and other numerical techniques to simulate the behavior of the system under different conditions. As a result, the computational costs of these methods can be high, requiring significant computing resources and time to obtain accurate results.

In comparison, machine learning can be a relatively inexpensive and faster method for designing hydrogen storage vessels. Machine learning techniques can analyze large datasets to identify patterns and relationships between various design parameters and the system's performance. These techniques can be used to optimize the design and predict the system's behavior under different conditions. However, machine learning models require a large amount of training data, and their accuracy may be limited by the quality of the data used for training.

Overall, while FEA, multi-scale modeling, and CFD are powerful tools for analyzing complex systems like hydrogen storage vessels, they can be computationally expensive and time-consuming. Machine learning, on the other hand, is a promising alternative that can potentially reduce the computational costs of the design process while maintaining the accuracy of the results. However, the optimal method will depend on the specific design requirements, available data, and computational resources.

3.) What can be calculated with respect to composite materials and hydrogen storage vessels ?

There are several calculations that can be made with respect to composite materials and hydrogen storage vessels, including:

Fiber volume fraction: This is the ratio of the volume of fibers to the total volume of the composite material. It is an essential parameter for determining the material's mechanical properties, such as its strength and stiffness.

Void content: This is the percentage of empty space within the composite material. Void content can affect the material's mechanical properties and its ability to store hydrogen.

Hydrogen storage capacity: This is the amount of hydrogen that can be stored in the composite material per unit mass or volume. It depends on the type of composite material used, its porosity, surface area, and other factors.

Hydrogen uptake/release kinetics: This refers to how quickly the composite material can absorb and release hydrogen. It is an important factor for determining the performance of the hydrogen storage vessel in practical applications.

Heat of adsorption: This is the amount of heat released or absorbed when hydrogen is adsorbed or desorbed from the composite material. It is important to understand the thermodynamics of hydrogen storage and to design effective hydrogen storage systems.

Cost analysis: This involves calculating the overall cost of manufacturing, testing, and operating a composite material-based hydrogen storage vessel, taking into account factors such as material cost, processing cost, and maintenance cost. Cost analysis can help determine the feasibility and commercial viability of using composite materials for hydrogen storage.

4.) How big are the errors of the model and numeric ?

The errors of the model and numerical methods used in the design of hydrogen storage tanks depend on several factors, such as the complexity of the system, the accuracy of the input data, and the assumptions made in the modeling process. Generally, more complex systems will have higher uncertainties and errors in the model and numerical predictions.

For example, in the case of hydrogen storage tanks, the errors of the model and numerical methods can arise from assumptions made regarding the tank's geometry, material properties, and boundary conditions. The accuracy of the input data used to model the system, such as the hydrogen properties and loading conditions, can also impact the accuracy of the predictions.

The errors associated with the numerical methods used in the design of hydrogen storage tanks can be quantified using techniques such as uncertainty analysis or sensitivity analysis. These techniques can provide insight into the sensitivity of the results to changes in the input parameters and the associated uncertainties.

5.) What is the difference between meso-macro and micro-meso-macro approaches ?

The meso-macro and micro-meso-macro approaches are two different methodologies used in materials science and engineering to understand the structure and behavior of materials. The main differences between the two approaches are:

Scale: The meso-macro approach focuses on the behavior of materials at the mesoscale, which is typically defined as the length scale between the microscale (atomic or molecular scale) and the macroscale (bulk or engineering scale). In contrast, the micro-meso-macro approach considers the behavior of materials across all three scales.

Modeling: The meso-macro approach typically uses continuum mechanics and finite element analysis to model the behavior of materials, whereas the micro-meso-macro approach employs a combination of atomistic simulations, mesoscale modeling, and continuum mechanics.

Applications: The meso-macro approach is often used to study the behavior of materials in large-scale engineering applications, such as aerospace structures or civil infrastructure. The micro-meso-macro approach, on the other hand, is commonly used in materials design and development, as well as in the study of complex material systems such as composites or biomaterials.

Complexity: The micro-meso-macro approach is generally more complex than the meso-macro approach due to the additional scales considered and the use of atomistic simulations. As a result, the micro-meso-macro approach can provide more detailed insights into the structure and behavior of materials, but also requires more computational resources and expertise.

In summary, the main difference between the meso-macro and micro-meso-macro approaches is the scale and modeling methods used to study materials. While the meso-macro approach is focused on the behavior of materials at the mesoscale and uses continuum mechanics, the micro-meso-macro approach considers all three scales and employs a combination of atomistic simulations, mesoscale modeling, and continuum mechanics.

  • Question 3

Comments on Section 6:

The authors should more clearly point out the following points:

1.) What are the objective functions, that are used in the optimization ?

In the case of optimizing a composite pressure vessel for hydrogen storage, the objective function is a mathematical expression that represents the performance of the composite material used in the vessel. The objective function is a critical component of the optimization process, as it defines the design goals and constraints that the optimization algorithm aims to satisfy. The objective function may include various design parameters, such as the material properties, vessel geometry, and operating conditions. For example, in the case of hydrogen storage vessels, the objective function may include factors such as the vessel's weight, storage capacity, and safety. The objective function must be carefully defined to ensure that it accurately represents the design goals and constraints. In some cases, local minima may exist, meaning that there may be multiple solutions that satisfy the optimization criteria. Therefore, it is important to consider multiple objective functions and perform sensitivity analysis to identify the optimal solution.

Optimization techniques such as gradient-based optimization or genetic algorithms can be used to search for the optimal solution that satisfies the objective function. These techniques iteratively adjust the design parameters until the objective function is optimized.

2.) Which parameters of the vessels are optimized ?

Several parameters of hydrogen storage vessels can be optimized to improve their performance and efficiency. These include:

  • Storage capacity: The amount of hydrogen that can be stored in the vessel per unit volume or mass can be optimized by adjusting the porosity, surface area, and material properties of the storage material.
  • Operating pressure: The pressure at which the vessel operates can be optimized to balance storage capacity, safety, and cost. Higher pressures can increase the storage capacity but may require stronger and more expensive materials.
  • Temperature: The operating temperature can affect the kinetics and thermodynamics of hydrogen uptake and release, and can be optimized to improve storage efficiency and performance.
  • Material selection: The choice of materials for the storage vessel can be optimized based on factors such as cost, durability, and safety. Composite materials, for example, can offer high storage capacity and strength, but may be more expensive than traditional materials.
  • Heat transfer: The efficiency of heat transfer in the storage vessel can be optimized to minimize heat losses and ensure consistent performance.
  • Design: The design of the storage vessel can be optimized to improve safety, durability, and ease of use. For example, the vessel may be designed to resist damage from impacts or vibrations, or to allow for easy filling and discharge of hydrogen.

Optimizing these parameters can lead to more efficient and effective hydrogen storage vessels, which can help accelerate the adoption of hydrogen as a clean and sustainable energy source.

3.) Which successes have been done in the design of the vessels using AI ?

The use of artificial intelligence (AI) in the design of hydrogen storage vessels is a relatively new field, but there have already been some notable successes. Some examples of these successes include:

  • Optimizing material properties: AI algorithms can be used to identify the most promising materials for hydrogen storage based on their properties and performance in simulations. This approach has been used to identify new metal-organic frameworks (MOFs) with high hydrogen storage capacity.
  • Designing new materials: AI algorithms can also be used to generate new materials with desirable properties for hydrogen storage. For example, researchers have used AI to generate new types of porous carbon materials with high surface area and stability, which are promising for hydrogen storage.
  • Accelerating materials discovery: AI can help accelerate the discovery of new hydrogen storage materials by automating the screening and analysis of large materials databases. For example, AI algorithms have been used to predict the properties of over 12,000 MOFs for hydrogen storage.
  • Improving system design: AI can be used to optimize the design of hydrogen storage systems, taking into account factors such as material properties, operating conditions, and safety. This can help improve the efficiency and reliability of hydrogen storage systems, making them more practical for real-world applications.

  • Question 4

Additional comments:

Most of the Figures seem to be copied from the literature.

A citation should be given in the caption of the figure.

As already mentioned above, the authors should check the situation concerning the copyrights of all Figures.

For example:

  • Figure 2 on page 4: reference [37] ?
  • Figure 3 on page 5: reference [38] ?
  • Figure 7 on page 9: reference ???
  • Figure 10 on page 12: reference [67] ?
  • Figure 11 on page 13: reference ???
  • Figure 12 on page 14: reference ???
  • Figure 14 on page 16: reference ???
  • Figure 17 on page 19: reference ???
  • Figure 18 on page 20: reference ???
  • Figure 19 on page 21: reference ???
  • Figure 21 on page 22: reference ???
  • Answer 4

Thank you for your feedback regarding our manuscript. We appreciate your attention to detail and agree that it is important to properly cite any figures that have been taken from previously published literature. We have carefully reviewed our manuscript and found that some of the figures were indeed taken from previous publications without proper citation in the figure captions. We apologize for this oversight and have made the necessary revisions to ensure that all figures are appropriately cited. We understand the importance of proper citation in scientific publications and strive to uphold the highest standards of academic integrity.

  • Question 5

Last sentence on page 10:

"Some common materials used in the design of composite hydrogen storage vessels include:"

The text continues with Figure 8

  • Answer 5

Thank you for this comment. The correction has been made in the revised version.

  • Question 6

page 12, line 400:

"Recently, the researchers investigate ... "

Which researchers? - please add a citation.

  • Answer 6

Thank you for this comment. The correction has been made in the revised version.

  • Question 7

pages 15 - 17:

The abbreviation for finite element analysis (FEA) is introduced 5 times on the pages 15 - 17.

This can be reduced to one time.

  • Answer 7

Thank you for this comment. The correction has been made in the revised version.

  • Question 8

On page 16, line 551:

"Another researcher, the authors reviewed existing ..."

Please add some name and reference.

  • Answer 8

Thank you for this comment. The correction has been made in the revised version.

  • Question 9

On page 18:

references for ABAQUS and ANSYS

  • Answer 9

Thank you for this comment. The correction has been made in the revised version.

  • Question 10

On page 19 (line 606):

What is the failure criterion?

The failure criterion of composite materials is a set of guidelines or equations that define the conditions under which a composite material will fail under different loading conditions. Failure in composite materials can occur in various forms, including matrix cracking, fiber breakage, delamination, and fiber-matrix debonding.

Generally, the failure criteria in composites are divided into two major groups: independent failure modes criteria (such as Tsai-Wu and Tsai-Hill) and dependent failure modes criteria (such as Hashin and Puck). It is noteworthy that several approximations could be introduced by independent failure mode criteria, as the inhomogeneity in materials (which governs failure mechanisms) is not considered in these criteria.

The correction has been made in the revised version.

  • Question 11

On page 26 (lines 832 - 834):

"In addition to machine learning algorithms, researchers ... "

Which researchers? please give a reference.

  • Answer 11

Thank you for this comment. The correction has been made in the revised version.

In the end, we are thankful to you for giving us the opportunity to improve our manuscript. We fully accept reviewer's valuable comments. At the same time, if you have any comments again, please do not hesitate to contact us.

Pr. Mourad NACHTANE

S Vertical Company, Paris, France

[email protected]

Round 2

Reviewer 1 Report

The revisions meet all my expectations. It is good work. It can be published without any further changes. Thanks. 

Reviewer 2 Report

The authors addressed the issues, which have been mentioned in the previous report.

The paper can be published in its current form.